



# Perspective on ice age Terminations from absolute chronologies provided by global speleothem records

Nikita Kaushal1, Carlos Perez-Mejias2, Heather M. Stoll3

1Department of Earth Sciences, ETH Zurich, Zurich, 80092, Switzerland; Now at the Earth and Planetary Sciences Department, American Museum of Natural History, New York, 10024, USA
2Institute of Global Environmental Change, Xi'an Jiaotong University, Xi'an, China
3Department of Earth Sciences, ETH Zurich, Zurich, 80092, Switzerland
*Correspondence to*: Nikita Kaushal

**Abstract.**

Glacial Terminations represent the largest amplitude climate changes of the last several million years. Several possible orbital-insolation triggers have been described to initiate and sustain glacial Terminations. Because of the availability of radiocarbon dating, the most recent Termination (TI) has been extensively characterized. Yet, it is widely discussed whether the sequence of feedbacks, millennial events and rates of change seen in TI is recurrent over previous Terminations. Beyond the limit of radiocarbon dating, records from the speleothem archive provide absolute age control through uranium-thorium dating and high-resolution proxy measurements. The PAGES SISALv3 global speleothem database allows us to synthesize the available speleothem records covering Terminations. However, speleothem climate signals are encoded in a number of proxies, and unlike proxies in other archives like ice or marine cores, the climatic interpretation of a given proxy can vary quite significantly among different regions. In this study, we

- synthesize the available speleothem records providing climate information for Terminations: TII, TIIIA,TIII, TIV and TV,
- present the records based on the aspect of climate encoded in the available records,
- examine the effects of different ice volume corrections on the final climate proxy record,
- evaluate whether there are leads and lags in the manifestation of Terminations across different aspects of the climate systems and different regions,
- we suggest directions for future speleothem research covering Terminations, speculate on suitable tuning targets among marine and ice core proxies, and discuss what model outputs maybe most suitable for comparison.

We find that TII has the greatest number of globally distributed records followed by TIIA and TIII. The records covering TIV and TV are largely restricted to the East Asian and Southeast Asian monsoon regions. Modelling and data-model comparison studies have greatly increased our understanding of the interpretation of oxygen isotope records across Terminations. Ice volume corrections have the most significant impact on European speleothem records with moisture sourced directly from the North Atlantic region. Within each Termination, a sequence of events can be established between a



sub-set of events and this sequence stays largely consistent across Terminations. However, improvements in dating and age-model uncertainties, higher resolution records and multi-proxy approaches are required to establish sequences within each sub-set of events. Beyond further research on targeted speleothem records, our recommendations for future directions include focusing on TII as a useful next target to understand climate dynamics, isotope-enabled transient simulations for better characterization of the other Terminations, and development of marine proxy records with signals common to speleothems to further improve the chronology of Terminations.

## 1 Introduction

Glacial Terminations represent the largest amplitude climate changes of the last several million years (Cheng et al., 2009, 2016b; Brook and Buizert, 2018). Over ~ 10 ky (10 kiloyear or 10,000 year) timescale, large northern hemisphere ice sheets retreat (e.g. Carlson and Winsor, 2012), and sea level rises (e.g. Gallup et al., 2002), and atmospheric $CO_2$ (Fischer et al., 1999) and global temperatures make a full transition from glacial to interglacial levels (e.g. Caillon et al., 2003). Several possible orbital-insolation triggers have been described to initiate and sustain glacial Terminations, and feedbacks between ice sheet retreat, ocean circulation and ocean carbon storage are invoked to explain the unstoppable progression (Mitsui et al., 2022; Bengtson et al., 2021; Capron et al., 2019 and references therein). Because of the availability of radiocarbon dating and the abundance of paleoclimate records from archives such as marine and ice cores, the most recent Termination (TI) has been the most extensively characterised (e.g. Stern and Lisiecki, 2014; Lamy et al., 2007; Denton et al., 2010; Lea et al., 2003; Barker and Knorr, 2021). Yet, it is widely discussed whether this sequence of feedbacks and millennial events, and rate of warming is recurrent over previous Terminations or is unique (e.g. Cheng et al., 2009; Barker and Knorr, 2021). Beyond the limit of radiocarbon dating, the chronologies of climate records from ice cores, deep ocean and lake sediments are often developed by tuning to orbital parameters which limits their use in understanding climate dynamics, particularly the response of the climate system to orbital forcing and the rates of response in the climate system (e.g. Lisiecki and Raymo, 2007; Raymo et al., 2006; Kawamura et al., 2007).

Speleothems provide absolute age control and high-resolution proxy measurements, with speleothems from a few cave regions covering multiple Terminations (Cheng et al., 2009, 2016b). In addition, the emerging field of band-counting speleothem confocal layers provides precise estimates of rates and durations in some settings (Stoll et al., 2022; Dong et al., 2022). The speleothem archive therefore provides unique records of climate change across Terminations, and additionally, may provide the opportunity to tune the chronology of ice and marine core archives. In order to do this, proxies for a common climate parameter or forcing have to be identified in both the speleothem and the marine/ice core records. For example, over TII, previous efforts have proposed a common regional sea surface temperature signal in marine records and a rainfall amount signal in speleothem $\delta^{18}O$ in western Italy (Drysdale et al., 2009), or a common surface ocean freshwater



signal in surface ocean $\delta^{18}O$ or $\delta^{18}O_{seawater}$ and speleothem $\delta^{18}O$ (Grant et al., 2012; Stoll et al., 2022). Long distance correlations between North Atlantic ice-rafted debris events and $\delta^{18}O$ anomalies in East Asian speleothem records have been proposed to tune North Atlantic marine records (e.g. Hobart et al., 2023). While such tuning approaches are promising, confidently extending speleothem tuning across multiple Terminations and precisely tuning millennial events across Terminations requires robust proxy interpretations across multiple archives and would be optimally further supported by modelling studies.

A challenge in speleothem records is that unlike proxies in other archives like marine or ice cores, the climatic interpretation of a given proxy can vary quite significantly among different regions. $\delta^{18}O$ is the most ubiquitously measured and interpreted proxy in speleothems. In addition to $\delta^{18}O$, speleothem climate signals are encoded in a number of other proxies like $\delta^{13}C$ and trace element ratios of Mg/Ca and Sr/Ca. Speleothem $\delta^{18}O$ records a very diverse set of climate influences in different environments, including: freshening of adjacent water bodies constituting the moisture source in the North Iberian and Black Sea regions (e.g Stoll et al., 2022; Badertscher et al., 2011), change in the dominant source of moisture in Northern Europe and North America (e.g. Cheng et al., 2016a, 2019), and change in temperatures in high altitude sites of Northern Europe and North America (e.g. Koltai et al., 2017; Lachniet et al., 2014), as well as aspects of the hydrological cycle including monsoon processes in Asia and South America (e.g. Cheng et al., 2013; Kathayat et al., 2016). Speleothem carbon isotopes are measured together with oxygen isotopes but are published less often and can have ambiguous interpretations, but in non-water limited mid-latitude regions may dominantly reflect temperature (Lechleitner et al., 2021; Stoll et al., 2023; Genty et al., 2001). In addition, the speleothem trace element proxy 'tool-box' has seen rapid evolution over the last two decades and provides valuable ways of independently reconstructing past climate and environmental parameters, as well as increasing our understanding of the signals encoded in the measured oxygen and carbon isotopes (Fairchild and Treble, 2009). For example, speleothem $\delta^{13}C$ values can further be impacted by the process of 'degassing' in the karst voids and the cave environment prior to the drip precipitating on the surface of the stalagmite (Fairchild et al., 2000). This is called prior calcite precipitation, and the speleothem trace element proxy Mg/Ca can be used to estimate the impact of prior calcite precipitation on speleothem $\delta^{13}C$ to reconstruct the temperature-sensitive 'initial' ('corrected') $\delta^{13}C$ of drip waters (Stoll et al., 2023).

There has been an ongoing discussion in the community that speleothem records with their chronological powers, and ubiquitously measured high resolution $\delta^{18}O$ proxy measurements, would be able to provide a more evolved understanding of the manifestation of Terminations in terrestrial regions, that are not covered by the more typical Termination records from marine and ice cores. In addition, the strength of the speleothem chronology could be extended to the marine and ice core records to better track forcings and responses during Terminations across the global climate system. However, so far, there has not been a systematic review of which speleothem records are available, their age control, or our current understanding of their climatic interpretations. There has also not been a targeted identification of gaps or most fruitful directions of future research to increase our understanding of Terminations using this speleothem-based approach. The Past Global Changes -

Speleothem Isotope Synthesis and AnaLysis i.e. PAGES-SISAL global speleothem database versions developed by the
PAGES-SISAL working group (Atsawawaranunt et al., 2018b; Comas-Bru et al., 2019, 2020b; Kaushal et al., 2024b;
Atsawawaranunt et al., 2018a, 2019; Comas-Bru et al., 2020a; Kaushal et al., 2024a) allow us to easily, and more
objectively, perform such a review.

In this review paper, we therefore synthesise the available speleothem records providing high resolution climate information
for Terminations: TII, TIIIA, TIII, TIV and TV. We first describe the aspect of the climate system encoded in records from
different regions and systems. For some records in which the ice volume effect on seawater $\delta^{18}$O is a significant portion of
the signal, we examine the options and effects of different ice volume corrections on the final climate proxy record. Then,
for each Termination, we evaluate whether there are consistent leads and lags in the manifestation of Terminations across
different aspects of the climate systems and different regions as tracked by speleothem proxy records. We briefly compare
the sequence of events across the different Terminations. And finally, considering the most up-to-date interpretation of what
climate signals are encoded by speleothem proxies in different regions, we provide future directions for speleothem records
covering Terminations, briefly speculate on suitable tuning targets among marine and ice core proxies, and discuss what
model outputs may be most suitable for comparison.

## 2 Stalagmite records and data processing

### 2.1 Criteria and compilation of records

Nearly all the speleothem records for this study have been taken from the latest PAGES-SISAL database version i.e.
SISALv3 database (Kaushal et al., 2024a, b). Since this Terminations project was undertaken parallel to building SISALv3
database, a special effort was made to ensure that records covering Terminations have been added to the database from
publications and repositories. Only single speleothem records have been used. We have not used any composite records in
this study.

Once SISALv3 was built, a search was made of SISALv3 for records covering the periods of interest, which generally
follow past work on Terminations by (Cheng et al., 2009) and (Cheng et al., 2016b), encompassing TII (140,000 to 120,000
y BP), TIIIA (230,000 to 210,000 y BP), TIII (258,000 to 238,000 y BP), TIV (350,000 to 330,000 y BP), and TV (440,000
to 420,000 y BP).

In representing records graphically, we first illustrate the caloric summer half-year insolation for each Termination following
(Tzedakis et al., 2017), and then we arrange the speleothem records in a series of geographical regions, namely, South
Europe, North Europe, North America, Central Asia, Indian Summer Monsoon (ISM), East Asian Summer Monsoon



(EASM), Southeast Asian Monsoon (SE Asia), and South American Monsoon. Rather than strict latitude-longitude boundaries, these geographical regions have been divided based on the dominant control on speleothem oxygen isotopes within loose latitude-longitude boundaries. South Europe (and the Middle East) include records with the $\delta^{18}O$ signal dominated by surface ocean $\delta^{18}O$ or the hydrological balance, contrasting with North Europe which mainly includes the
alpine records with strong temperature dependency of meteoric precipitation on oxygen isotopes. North American Lehman cave oxygen isotopes similarly record temperature dependency of meteoric precipitation while the Buckeye Creek cave records encode information of the dominant season (source) of precipitation that varies between glacial and interglacial times. The Central Asian records are located far inland and are influenced by the Westerlies and record supra-regional atmospheric circulation changes. The monsoon records in the ISM, EASM, SE Asian and South American regions are
largely driven by Rayleigh distillation effects. Section 3 covers the interpretation of oxygen isotopes in further detail. For European records, where robust carbon isotope interpretations are available, and provide additional information on Terminations, we also discuss the $\delta^{13}C$ and Mg/Ca records. While $\delta^{18}O$ and $\delta^{13}C$ provide information on the timing temperature change, absolute temperature change can only be reconstructed from other proxies such as fluid inclusions. Such proxy records are few in number, of lower resolution and are not yet included in the SISAL database versions. These records
are only briefly summarized in Section 3.3.

All records covering the periods of interest and within each geographical region were plotted and examined. Sub-aqueous speleothems have not been considered further due to the added uncertainty in their U-Th age control and oxygen isotopic interpretations (namely the Devil's Hole records and one of the Corchia cave records). Among the remaining records, the
main manuscript map (Fig. 1), time series plots (Fig. 2) and table (Table 1) prioritise the records that cover the longest time period, with the highest sampling resolution, with the largest number of U-Th dates, and lowest errors in age-depth models covering the period of interest. The different criteria listed above have been taken on balance and records that may be nearly as good have been given in the ssupplementary information figures and table. Remaining records that very partially cover the period of interest and/or are of very low resolution in a way that they give no additional information on the Termination have
not been considered further within the main manuscript although a few records have been plotted in supplementary information (Supp. Figs. 1, 2 and Supp. Table 1). Where author generated age-depth models have uncertainty measurements, these have been preferred. Where author generated age-depth models are not available in SISALv3, then SISAL generated age models in COPRA/Bchron/Bacon (in that order) have been used. Composite records such as Soreq_composite and PEK_composite do not have author generated age-depth model uncertainties and these cannot be
generated by automated scripts typically used by the SISAL working group. Such composite records have been shown in supplementary information figures. Although the SISAL working group have made an effort to provide standardised age-depth chronologies for all speleothems in the database, not all speleothems produced realisations for all age models (further details can be found in the related publications, see (Comas-Bru et al., 2020b; Amirnezhad-Mozhdehi and Comas-Bru, 2019; Rehfeld et al., 2020; Rehfeld and Bühler, 2024). In one case, the SE Asian Green Cathedral cave record (namely Meckler et



al., 2012), the age-depth model has been requested directly from the author, since this record proved to be challenging for dating, doesn't have author-generated age uncertainties in SISALv3, and is not amenable for standardised age-depth model creation. The SE Asian Abadi cave record was published after the SISALv3 database was closed for submissions, and has been added directly to this study. The Sanbao cave SB11 record $\delta^{18}$O data points between 228,000-226,000 y BP have been added from NCEI-NOAA, the remaining data points are taken from SISALv3. All speleothems, other than the Crovassa Azurra record, considered in this study are composed of calcite. Crovassa Azurra has a mixed mineralogy (dominantly composed of aragonite), where the mineralogy of each oxygen isotopic sub-sample has been provided in SISALv3, the stalagmite from this cave has been plotted in the supplementary information of this manuscript, and no change has been made to the oxygen isotopic measurement in response to the change in mineralogy. Under equilibrium conditions, aragonite is calculated to have $\delta^{18}$O values that are 0.8 per mille more enriched than calcite at 25$^{\circ}$ C (Kim et al., 2007).

**Table 1: Details of speleothem records covering Terminations.**

| region | site name | latitude | longitude | elevation (m) | entity name | age model | data source | citation | primary interpretation | secondary interpretation | comments |
|---|---|---|---|---|---|---|---|---|---|---|---|
| **Termination II** | | | | | | | | | | | |
| **South Europe** | Corchia | 43.9833 | 10.2167 | 840 | CC-5_2018 | SISAL-Bchron | SISALv3 | (Tzedakis et al., 2018) | rainfall amount | source water composition | |
| | La Vallina | 43.4100 | -4.8067 | 70 | Garth | author-mixed Bchron | SISALv3 | (Stoll et al., 2022) | source water composition | temperature dependency of meteoric precipitation | confocal band counting |
| **North Europe** | Abaliget | 46.1333 | 18.1167 | 209 | ABA_1 | author-StalAge | SISALv3 | (Koltai et al., 2017) | temperature dependency of meteoric precipitation | source water composition | flowstone; author generated age model uses both cores to create master chronology |





| | | | | | | | | | | |
|---|---|---|---|---|---|---|---|---|---|---|
| **North America** | Lehman caves | 39.0100 | -114.2200 | 2080 | LMC-14 | SISAL-copRa | SISALv3 | (Lachniet et al., 2014) | temperature dependency of meteoric precipitation | change in moisture source latitude | |
| | | | | | LMC-21 | SISAL-copRa | SISALv3 | (Lachniet et al., 2014) | temperature dependency of meteoric precipitation | change in moisture source latitude | |
| | | | | | LC-2 | SISAL-Bchron | SISALv3 | (Shakun et al., 2011) | | | |
| **Central Asia** | Tonnelnaya | 38.4000 | 67.2300 | 3226 | TON-1 | SISAL-copRa | SISALv3 | (Cheng et al., 2016a) | large-scale circulation and supra-regional climate | change in moisture transport trajectory and linked seasonality | temperature effect on calcite precipitation |
| **ISM** | Bittoo | 30.7903 | 77.7764 | 3000 | BT-2.3 | SISAL-copRa | SISALv3 | (Kathayat et al., 2016) | large-scale circulation; moisture transport history | | |
| | Xiaobailong | 24.2000 | 103.3600 | 1500 | XBL-26 | SISAL-copRa | SISALv3 | (Cai et al., 2015) | rainfall amount | | |
| **EASM** | Dongge | 25.2833 | 108.0833 | 680 | D4_2005_Kelly | author-unknow | SISALv3 | (Kelly et al., 2006) | change in seasonality | upstream rainout | |





| | | | | | | | | | | |
|---|---|---|---|---|---|---|---|---|---|---|
| | | | | | | n | | | | |
| | Hulu | 32.5000 | 119.1700 | 86 | MSX | SISAL-BChron | SISALv3 | (Cheng et al., 2006) | oxygen isotopes of meteoric precipitation | summer to winter precipitation intensity | |
| **South America** | Diamante | -5.7300 | -77.5000 | 960 | NAR-C | sisal-Bchron | SISALv3 | (Cheng et al., 2013) | rainfall amount | upstream rainout | Amazon recycled moisture more positive than distal ocean source |
| **Termination IIIA** | | | | | | | | | | | |
| **South Europe** | Sofular | 41.4167 | 31.9333 | 440 | SO-4 | SISAL-Bchron | SISALv3 | (Badertscher et al., 2011) | source water composition | | more negative from Caspian Sea with melt waters, more positive from Mediterranean |
| **North Europe** | Spannagel | 47.0800 | 11.6700 | 2310 | SPA121_2021 | author-OxcCal | SISALv3 | (Wendt et al., 2021) | local winter temperature | change in moisture transport trajectory | source water composition indicated by a possible short melt water pulse excursion |
| **North America** | Buckeye Creek | 37.9800 | -80.4000 | 600 | BCC-9 | SISAL-copRa | SISALv3 | (Cheng et al., 2019) | change in seasonality | | |



| | | | | | | | | | large-scale circulation and supra-regional climate | change in moisture transport trajectory and linked seasonality | temperature effect on calcite precipitation |
|---|---|---|---|---|---|---|---|---|---|---|---|
| **Central Asia** | Kesang | 42.8700 | 81.7500 | 2000 | KS06-A | author-linear between dates | SISALv3 | (Cheng et al., 2016a) | | | |
| **EASM** | Dongge | 25.2833 | 108.0833 | 680 | D8 | SISAL-copRa | SISALv3 | (Kelly et al., 2006) | change in seasonality | upstream rainout | |
| | Sanbao | 31.6670 | 110.4333 | 1900 | SB11 | 228-226 ka author-linear between dates; remaining SISAL-copRa | 228-226 ka NCEI-NOAA; remaining SISALv3 | (Cheng et al., 2016b) | upstream rainout | change in seasonality | |
| | | | | | SB61 | author-linear between dates | SISALv3 | (Cheng et al., 2016b) | upstream rainout | change in seasonality | |
| **SE Asia** | Abadi | -5.0000 | 119.7000 | 300 | AC-11-06 | author-Bacon and tie points | publication | (Kimbrough et al., 2023) | rainfall amount | | |
| **South America** | Diamante | -5.7300 | -77.5000 | 960 | NAR-D | SISAL-copRa | SISALv3 | (Cheng et al., 2013) | rainfall amount | upstream rainout | Amazon recycled moisture more positive than distal ocean source |



| | | | | | | | | rainfall amount | upstream rainout | Amazon recycled moisture more positive than distal ocean source |
|---|---|---|---|---|---|---|---|---|---|---|
| | | | | | NAR-F | SISAL-Bchron | SISALv3 | (Cheng et al., 2013) | | |
| **Termination III** | | | | | | | | | | |
| **South Europe** | Ejulve | 40.7600 | -0.5900 | 1240 | ARTEMISA | author-mixed StalAge | SISALv3 | (Pérez-Mejías et al., 2017) | temperature change: not specified | source water composition | record focuses on $\delta^{13}$C as proxy for dry conditions |
| **North Europe** | Spannagel | 47.0800 | 11.6700 | 2310 | SPA121_2021 | author-OxcCal | SISALv3 | (Wendt et al., 2021) | local winter temperature | change in moisture transport trajectory | source water composition indicated by a possible short melt water pulse excursion |
| **North America** | Buckeye Creek | 37.9800 | -80.4000 | 600 | BCC-9 | SISAL-copRa | SISALv3 | (Cheng et al., 2019) | change in seasonality | | |
| **Central Asia** | Kesang | 42.8700 | 81.7500 | 2000 | KS08-1 | SISAL-copRa | SISALv3 | (Cheng et al., 2016a) | large-scale circulation and supra-regional climate | change in moisture transport trajectory and linked seasonality | temperature effect on calcite precipitation |
| **ISM** | Xiaobailong | 24.2000 | 103.3600 | 1500 | XBL-26 | SISAL-copRa | SISALv3 | (Cai et al., 2015) | rainfall amount | | |





| | | | | | | | | | | |
|---|---|---|---|---|---|---|---|---|---|---|
| **EASM** | Sanbao | 31.6670 | 110.4333 | 1900 | SB61 | SISAL-copRa | SISALv3 | (Cheng et al., 2016b) | upstream rainout | change in seasonality | |
| **SE Asia** | Green Cathedral | 4.2333 | 114.9250 | 569 | GC08 | author-StalAge | author | (Meckler et al., 2012) | source composition; change in pathway, regional rainfall amount | land exposure | large U/Th error associated with open conditions; ITCZ moving South |

### Termination IV

| | | | | | | | | | | |
|---|---|---|---|---|---|---|---|---|---|---|
| **EASM** | Sanbao | 31.6670 | 110.4333 | 1900 | SB61 | author-linear between dates | SISALv3 | (Cheng et al., 2016b) | upstream rainout | change in seasonality | |
| **SE Asia** | Abadi | -5.0000 | 119.7000 | 300 | AC-09-04 | author-Bacon and tie points | publication | (Kimbrough et al., 2023) | rainfall amount | | |

### Termination V

| | | | | | | | | | | |
|---|---|---|---|---|---|---|---|---|---|---|
| **EASM** | Sanbao | 31.6670 | 110.4333 | 1900 | SB-14 | SISAL-Bacon | SISALv3 | (Cheng et al., 2016b) | upstream rainout | change in seasonality | |



| | | | | | | | | | | | |
|---|---|---|---|---|---|---|---|---|---|---|---|
| **Southeast Asia** | Green Cathedral | 4.2333 | 114.9250 | 569 | GC08 | author-StalAge | author | (Meckler et al., 2012) | source composition; change in pathway, regional rainfall amount | land exposure | large U/Th error associated with open conditions; ITCZ moving South |








**Figure 1: Maps showing locations of cave sites with speleothem records covering Terminations II, IIIA, III, IV and V respectively.**









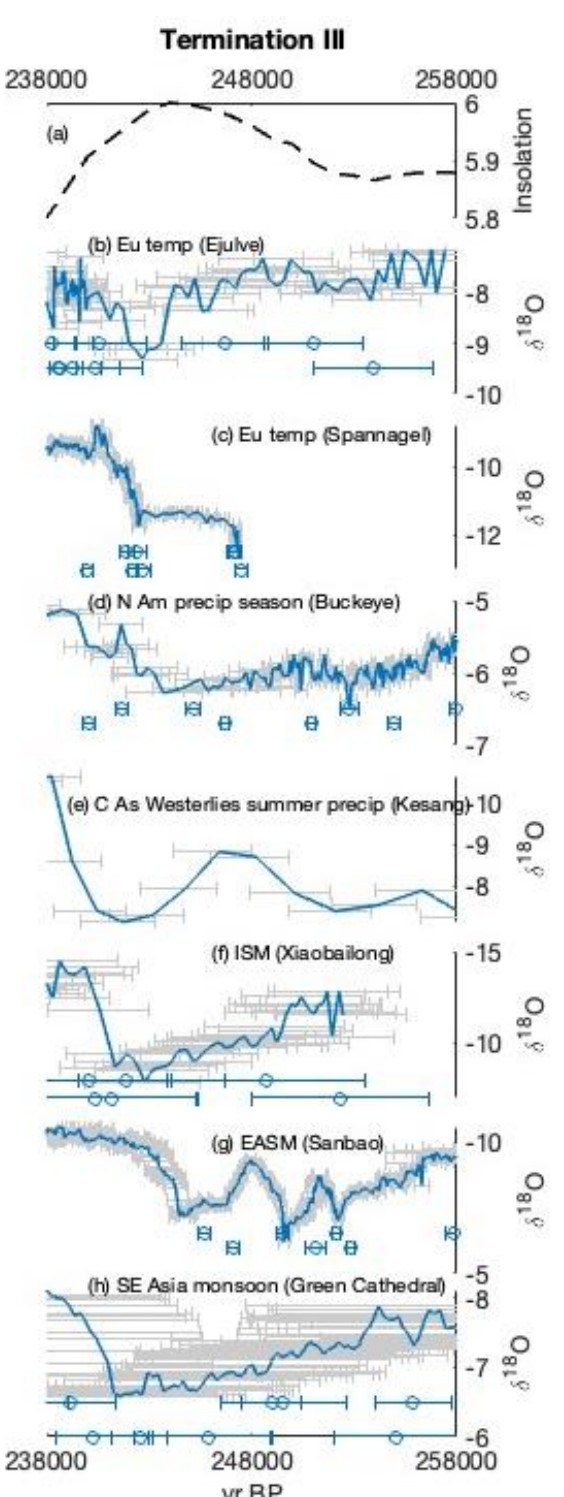

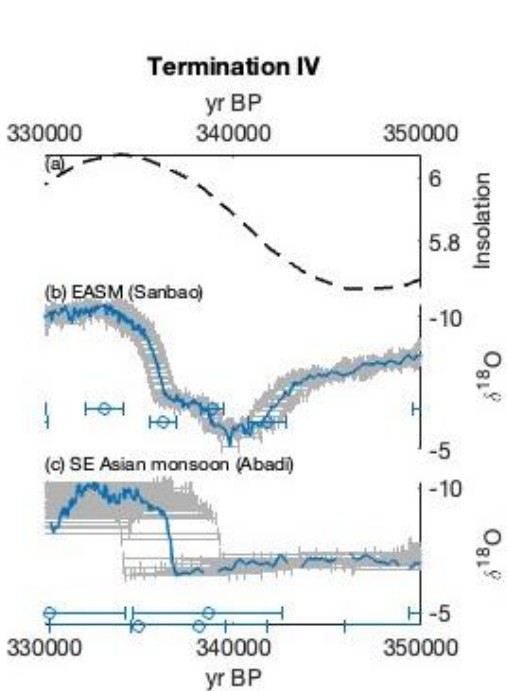



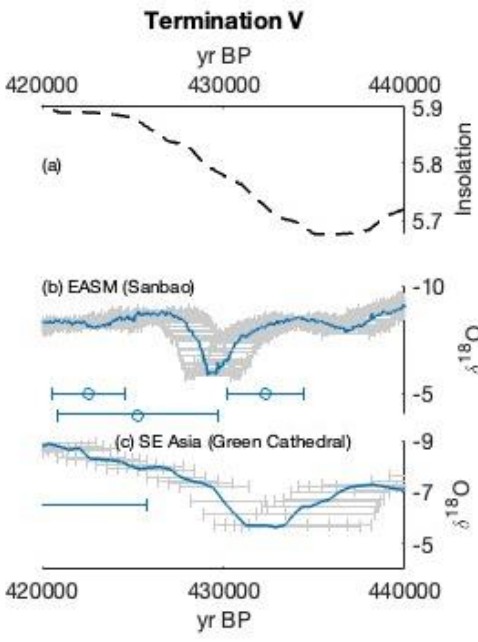


**Figure 2: Ages covering Terminations are plotted against oxygen isotopic measurements. Age-depth model uncertainties, U-Th dates and dating uncertainties have also been plotted. Insolation curve is the summer half year caloric insolation as provided in Tzedakis et al, 2017. Further record information can be found in Table 1. Record locations are plotted in the maps in Figure 1. [precip = precipitation; temp = temperature; N Eu = North Europe; S Eu = South Europe; N Am = North America; C As =**
**Central Asia; ISM = Indian Summer Monsoon; EASM = East Asian Summer Monsoon; SE Asia = Southeast Asia; S Am = South America]**

### 2.2 Accounting for ice volume effects in speleothem oxygen isotope records

The shrinking of global ice sheets across glacial Terminations leads to the release of the isotopically negative freshwater in ice sheets back into the ocean, leading to a negative shift in the average $\delta^{18}O_{seawater}$ estimated at 1 to 1.3 ‰ (Duplessy et al.,
2002; Adkins et al., 2002; Schrag et al., 2002, 1996). Most of the glacial meltwater during Terminations is released by North Atlantic ice sheets directly into the surface North Atlantic ocean and adjacent marginal seas, and is subsequently mixed into the deep North Atlantic by convection and circulated throughout the global surface and deep ocean (Jackson and Wood, 2018). Radiocarbon tracers and modelling of the last deglaciation suggest the deglacial $\delta^{18}O_{seawater}$ signal reaches the deep and surface Pacific ocean with a 1,000 to 3,000 year lag (Duplessy et al., 1991).


Because the surface ocean is the ultimate source for all water precipitated on land and infiltrating in caves, this change in the initial isotopic composition of the surface ocean during glacial Terminations is transferred to the drip water and precipitated stalagmites (e.g. Stoll et al., 2022). In some regions, other sources of $\delta^{18}O$ variability significantly exceed the magnitude of the global ice volume effect and it can be neglected for interpretation. However, the ice volume component can be a
significant component of the $\delta^{18}O$ variation in some speleothem records, and it can be helpful to isolate the non-ice volume





components to clarify interpretations. Therefore we explore the consequences and uncertainties surrounding the ice volume correction.

In order to examine the effects of this ice volume change on speleothem $\delta^{18}O$ Termination records, a value of 0.00833 ‰ / m of relative sea level has been used to calculate the global average change in $\delta^{18}O_{seawater}$ with the progressive melting of ice sheets (Adkins et al., 2002; Elderfield et al., 2012; Shakun et al., 2015). For TII through TV, past sea level estimations are used from two types of sources. In one source type, the sea level component has been estimated from benthic $\delta^{18}O$ and its chronology is based on orbital tuning; for this type, we employ the long duration curve of Spratt and Lisiecki (2016) for all Terminations. The Spratt & Lisiecki solution is proposed to have an uncertainty of 4,000 years in the Late Pleistocene. As a

second source type, we also employ a sea level curve derived from the sensitivity of Red Sea salinity to the water depth at the sill where inflow occurs. The Red Sea sea level features a chronology tuned by correlating millennial dust events in Red Sea to $\delta^{18}O$ anomalies in East Asian monsoon speleothem records (Grant et al., 2014). These two records exhibit some differences in the amplitude and timing of sea level rise during glacial Terminations (Supp. Fig. 3). A further benthic $\delta^{18}O$-derived curve (Waelbroeck et al., 2002) has a lower resolution than the other curves and is shown in Supplementary Figure 3

but not used for correction of the speleothem records. We employ the uncertainties from the original authors, which define a range of sea levels for a given age. Thus, 'ice volume-corrected' speleothem $\delta^{18}O$ records have an additional uncertainty in the isotope trend due to the uncertainties of the chronology from the ice volume curve.

In addition to the globally distributed ice volume effect, changes in regional hydrological balance of evaporation over

precipitation may generate additional increases or decreases in the $\delta^{18}O_{seawater}$ of the surface ocean. For example, over the last glacial Termination, paired Mg/Ca and $\delta^{18}O_{foram}$ measurements indicate that the $\delta^{18}O_{seawater}$ of the surface North Atlantic ocean experienced a larger amplitude change than the global average ocean (Waelbroeck et al., 2014). Additionally, during the early stages of deglaciation the $\delta^{18}O_{seawater}$ of the North Atlantic changes much more than the global surface ocean because the North Atlantic directly receives meltwater from Northern Hemisphere ice sheets. For inland European speleothems in which

the $\delta^{18}O$ signal is interpreted as temperature record, and for which precipitation has a North Atlantic moisture source, it is ideal to correct for the full regional change in source water in order to deconvolve this effect from the temperature signal. At present this is possible for TII, by subtracting the $\delta^{18}O$ from a coastal North Iberian Speleothem Archive (NISA) record, whose signal is dominated by the regional evolution of $\delta^{18}O_{seawater}$ of the North Atlantic (see Stoll et al., 2022). In other regions, paired foraminiferal Mg/Ca and $\delta^{18}O_{calcite}$ measurements indicate that variation in the $\delta^{18}O_{seawater}$ may be larger or smaller than the

globally averaged ice volume effect (Waelbroeck et al., 2014), depending on factors such as the Inter-Tropical Convergence Zone (ITCZ) position. We do not explicitly adjust for these variations since they generally amplify, rather than obscure, the hydrological signal present in the stalagmite $\delta^{18}O$.



Wherever correction for changing $\delta^{18}O_{seawater}$ is implemented, the speleothem data have been binned to 1000, 250 and 125 years
respectively to accommodate uncertainties in the speleothem chronology (Supp. Fig. 4). The change in $\delta^{18}O_{seawater}$ as a result of
freshening has simply been subtracted from the speleothem $\delta^{18}O$ in these bins. Since the uncertainty on sea level curves is
much greater than the uncertainty on speleothem age-depth models, only the uncertainty on sea level curves has been
considered in these plots.

### 2.3 Degassing corrections for speleothem carbon-isotope based temperature records

The speleothem $\delta^{13}C$ record from some mid-latitude regions can also be used for temperature reconstructions (e.g. Held et al,
2024). The temperature sensitivity is well documented over TI, where in coastal Northwest Iberia the NISA $\delta^{13}C$ records vary
inversely with regional temperature records from marine archives (Stoll et al., 2023). Multiproxy process modelling suggests
that the correlation arises because higher temperatures increase vegetation productivity in this region, enhancing soil $CO_2$
production as well as an oversupply of $CO_2$ to karst waters, both of which result in more negative speleothem $\delta^{13}C$ values
(Lechleitner et al., 2021; Stoll et al., 2023, 2022). Speleothem $\delta^{13}C$ values can further be impacted by the process of
'degassing' in the karst and the cave environment prior to the drip precipitating on the surface of the stalagmite (Johnson et
al., 2006; Lechleitner et al., 2021). This is called prior calcite precipitation (PCP) (Fairchild et al., 2000). Variations in the
extent of PCP during the growth of a speleothem can alter the trends in the speleothem $\delta^{13}C$. Fortunately, the speleothem
Mg/Ca ratio can be used to quantify changes in the fraction of dripwater Ca which has been lost to PCP within a given
speleothem. For example, intervals of high Mg/Ca ratio imply a more significant Ca loss and PCP, entailing a positive shift
in the speleothem $\delta^{13}C$. Using speleothem Mg/Ca it is now possible to quantitatively remove the impact of variable PCP on
speleothem $\delta^{13}C$ time series to reconstruct the temperature-sensitive initial variations in $\delta^{13}C$ of dripwaters (Stoll et al., 2023).
Since the $\delta^{13}C$ of dripwater acquired through the equilibration of rain water with soil gas and bedrock dissolution is what
encodes the temperature signal and is of primary interest to us, we here 'correct' for the additional karst and in-cave
influence of degassing and PCP on the $\delta^{13}C$ value by plotting the $\delta^{13}C_{corr}$ for select records (Genty et al., 2003, 2006) (Supp.
Fig. 5). Further method details can be found in Stoll et al., 2023.

### 3 Speleothem records of diverse climate aspects over Terminations

We review the climatic processes which are encoded in oxygen isotopic ratios of speleothems from different regions.

### 3.1 Records of surface ocean freshening

Changes in the $\delta^{18}O_{seawater}$ of the moisture source for caves and drip waters may be the dominant signal in speleothem $\delta^{18}O$ in
some settings. The $\delta^{18}O$ in speleothems from coastal caves in Northwest Spain (NISA) is dominantly controlled by the
$\delta^{18}O_{seawater}$ of the eastern North Atlantic, as documented in comparison with independently dated $\delta^{18}O_{seawater}$ records from
foraminifera over TI (Stoll et al., 2022). Because of its proximity to the source of meltwater release, the $\delta^{18}O_{seawater}$ of the



surface ocean in the North Atlantic experiences a higher amplitude change in $\delta^{18}O_{seawater}$ across a glacial cycle, and may record
transient millennial scale events in the $\delta^{18}O_{seawater}$. Over TII, NISA speleothems provide a record of the timing of deglacial
freshening of the eastern North Atlantic with a $\delta^{18}O$ amplitude of ~2.5 ‰ (Supp. Fig. 6). Other coastal caves on the Atlantic
margin, such as Villars Cave (Supp. Fig. 2), may also be dominated by the change in isotopic composition of the North
Atlantic.

Similarly, the $\delta^{18}O$ in speleothems from Sofular Cave, bordering the Black Sea in Northern Turkey, is inferred to be most
strongly influenced by the $\delta^{18}O_{seawater}$ of the Black Sea surface waters. The Black Sea is proposed to receive direct meltwater
runoff from the Southern Margin of the Eurasian ice sheet, leading to depleted surface water $\delta^{18}O$ during periods of Eurasian
ice sheet retreat with up to 10 ‰ freshening signal in the surface of the Black Sea (Wegwerth et al., 2019; Held et al.,
2024).

In addition to ice volume meltwater effects, in some settings, changes in rainfall patterns may also lead to changes in the
$\delta^{18}O_{seawater}$ of the surface ocean moisture source. Increased local or regional precipitation relative to evaporation may lead to
freshening and more negative $\delta^{18}O_{seawater}$ of surface waters. This effect may potentially augment the tendency for hydrological
fractionation to generate more negative rainfall $\delta^{18}O$ with higher rainfall amounts (e.g. Breitenbach et al., 2010).

### 3.2 Temperature-sensitive records from oxygen and carbon isotopes

The $\delta^{18}O$ of precipitation decreases appreciably by Rayleigh distillation with progressive rainout as moisture-laden air masses
move inland and to higher altitudes (Dansgaard, 1954, 1964; Alley and Cuffey, 2018). Colder temperatures both increase the
fractionation and accelerate rainout, so colder temperatures lead to decrease in the precipitation $\delta^{18}O$ at inland, high altitude,
and high latitude locations in regions with mean annual temperatures lower than about 10°C (Harmon et al., 1979; Rozanski
et al., 2013; Denton et al., 2005). Speleothem $\delta^{18}O$ records from these regions provide information on the timing of
temperature change, but not absolute temperatures.

In the European Alps and Iberian Range, this temperature control on Rayleigh distillation leads to a dominant temperature
effect on speleothem $\delta^{18}O$ producing anomalies of ~ 3 ‰ (Supp. Fig. 7). Consequently, records from central Europe (Sieben
Hengste, Schneckenloch, Schafsloch, Holloch im Mahdtal, Abaliget, Spannagel) and the Iberian Range (Ejulve) exhibit
lowest $\delta^{18}O$ during coldest temperatures. At the same time, since the North Atlantic is a dominant moisture source for most
central European caves (Affolter et al., 2020), the magnitude of regional North Atlantic $\delta^{18}O_{seawater}$ change over Terminations is
also imprinted on these records. For TII, we illustrate the signal when this regional $\delta^{18}O_{seawater}$ component is subtracted from





these records using the NISA coastal Iberian record (Supp. Fig. 4). In contrast to the original $\delta^{18}O$ which featured an apparent warm signal during TII, then cooling prior to late deglacial warming, the corrected Abaliget record features a stable cold glacial period until the final deglacial warming. The early deglacial warming in the Schneckenloch record is amplified by this correction. We further refer to these regional source-corrected trends in Section 4.1. The temperature sensitive La

Vallina cave Garth $\delta^{13}C_{corr}$ record from the NISA coastal Iberian record, supports a two-step warming (135 ka and 129 ka respectively) where the older warming step has no corresponding data from the corrected Abaliget record but the younger warming step lines up with the Abaliget record. The absence of an equivalent absolute dated record of North Atlantic $\delta^{18}O_{seawater}$ evolution in prior Terminations precludes regional correction of temperature equivalent European records in TIII or older at this time, instead the global ice volume correction has been applied to these records. Similar to TII, the TIII Ejulve

cave temperature sensitive $\delta^{13}C_{corr}$ record from the Iberian region extends the available temperature data for this Termination from Europe.

Where multiple ocean moisture sources with different rainout extents contribute to precipitation over a cave site, changes in the relative significance of the two moisture sources may influence the $\delta^{18}O$ of precipitation in addition to the temperature

control. This has been proposed for the joint Mediterranean and Atlantic sources to the Iberian Range cave systems (Pérez-Mejías et al., 2017).

In western North America, the $\delta^{18}O$ record of Lehman Cave (Nevada) is also most sensitive to regional temperatures producing a $\delta^{18}O$ anomaly of ~2 ‰. Isotope-enabled General Circulation Model (GCM) simulations of precipitation $\delta^{18}O$ at

the Last Glacial Maxima (LGM) and preindustrial reveal a net decrease in $\delta^{18}O_{precip}$ inland at the LGM, compensating for higher glacial $\delta^{18}O$ of surface seawater. This is dominantly due to greater rainout from stronger uplift and cooling, with a secondary effect of greater cool season precipitation influenced by ice sheet topography and cooling of Pacific sea surface temperatures (Tabor et al., 2021). Given the Pacific moisture source, the global ice volume correction is the relevant one for accounting for isotopic changes in the ocean moisture source region, but this correction has minimal effect on the trends.

**3.3 Other temperature records**

Fluid inclusion hydrogen and oxygen isotopes, clumped isotope thermometry, ratios of Glycerol Dialkyl Glycerol Tetraethers (GDGTs) such as TEX86, noble gas content, and homogenization of fluid inclusions all have been employed for temperature reconstructions. Calibrations to absolute temperature exist for these methods based on modern observations.

However, the larger sample sizes and/or more challenging analytical routines mean fewer data points can be measured which often additionally have reasonably large errors, and the data from these proxies are not yet available in the SISAL





speleothem database versions. Consequently, these records have been synthesized in a map (Fig. 3) and a table (Table 2) rather than plotting on time series axes.

Over TII, these records include the Schrattenkarst (Wilcox et al, 2020) and the Hungarian caves temperature reconstruction (Demény et al., 2021) using fluid inclusion hydrogen isotopes, which suggests a temperature increase of 4.8 (+/- 1.9) and 7.4 (+/- 1.6) ⁰C respectively. The Villars cave record which is located much closer to the coast suggests a higher temperature increase of 15 (+/- 1.9) ⁰C using a combination of fluid inclusion oxygen isotopes and clumped isotope thermometry (Wainer et al., 2011). The South Europe, Soreq (Matthews et al., 2021) and Pentadactylos (Nehme et al., 2020) temperature

records have similarly been reconstructed using fluid inclusion oxygen isotopes and clumped isotope thermometry and show similarly higher temperature increases of 10 (+/- 2) and 15⁰C (+/- 2.7) respectively over TII. The Jiangjun cave record from the EASM region has been reconstructed using TEX86 and shows a moderate temperature increase of 4.1 (+/- 0.5) ⁰C (Wassenburg et al., 2021). For this Termination, the records suggest that temperature increase is highest in speleothem records along the Atlantic and Mediterranean coasts and the temperature progressively decreases into the Alps, and into the

EASM region.

For the older Terminations, very few such speleothem temperature records are available. The Hungarian caves study (Demény et al., 2021) suggests that unlike TII there was limited temperature variability during TIII. The SE Asian Whiterock cave temperature study (Meckler et al., 2015) compares a number of different reconstruction methods (see Table

2) to provide a moderate temperature increase of 3.5 (+/- 1)⁰C over TV.

**Table 2: Table giving information of speleothem temperature records covering Terminations using methods such as fluid inclusion hydrogen and oxygen isotopes, clumped isotope thermometry, ratios of Glycerol Dialkyl Glycerol Tetraethers (GDGTs) such as TEX86, noble gas content, and homogenization of fluid inclusions.**


| Site name | Entity name | Termination | Glacial temperature | Interglacial temperature | Method | Citation | Comment |
|---|---|---|---|---|---|---|---|
| Hungarian caves (Abaliget, Baradla, Beke, Ajandek) | ZEP, BAR-II, BNT-2, AJ-1, ABA-1, ABA-2 | TII | 2.8 (+/- 1.6) | 10.2 (+/- 1.6) | Fluid inclusion hydrogen isotopes | (Demény et al., 2021) | For this study, 3 lowest and 3 highest temperatures in the time period from 140000 to 120000 were averaged to give the glacial and interglacial temperatures respectively. |
| | | TIII | 7.5 (+/- 1.6) | | | | Little temperature variability between 250 ky BP to 230 ky |





| | | | | | | | |
|---|---|---|---|---|---|---|---|
| | | | | | | | BP. The average of all measurements is given here. |
| Schrattenkarst | M37-1-23A, M37-1-16C, M37-1-16A, M6-73-3 | TII | 1.5 (+/- 0.5) | 5.8(+/-1.4) | Fluid inclusion hydrogen isotopes | (Wilcox et al., 2020) | Glacial temperature estimated from Fig. 2 in the publication and taken as the one at 131 ky BP. Interglacial temperature calculated based on value above glacial as stated in the publication. |
| Villars | Vil-car-1 | TII | 5.5 or 7.9(+/-1.9) | 18.9 or 22.9 (+/- 1.9) | Fluid inclusion oxygen isotopes, clumped isotopes and modern observations. | (Wainer et al., 2011) | Values as given in the conclusion of the paper. |
| Soreq | Soreq speleothems | TII | 10-12(+/-2) | 20-22(+/-2) | Fluid inclusion oxygen isotopes and clumped isotope thermometry. | (Matthews et al., 2021) | Values as given in the abstract and conclusion of the paper. Errors estimated from Fig. 8b of the paper. |
| Pentadactylos | Pentadactylos (Penta-1 and Penta-2) | TII | 14.7(+/-2.7) | 30.6(+/-2.7) | Fluid inclusion oxygen isotopes and clumped isotope thermometry. | (Nehme et al., 2020) | Coldest values from ~141-127 ky BP as given in the publication considered to be glacial values. Value for 127 ka as given in the publication considered to be the interglacial value. |
| Jiangjun | JJ0403, JJ0406 | TII | 15.7(+/-0.5) | 19.8(+/-0.5) | TEX86 | (Wassenburg et al., 2021) | For this study, 3 lowest and 3 highest temperatures in the time period from 140 to 120 ky BP were averaged to give the glacial and interglacial temperatures respectively. |
| Whiterock | WR5 | TV | 19(+/-1) | 22.5(+/-1) | Average of several methods: noble gas, clumped isotope, homogenization temperature, fluid inclusion oxygen, and fluid inclusion hydrogen isotopes | (Meckler et al., 2015) | Temperatures and errors for the period from 440 to 420 ky BP estimated from Fig. 9 of the paper. |



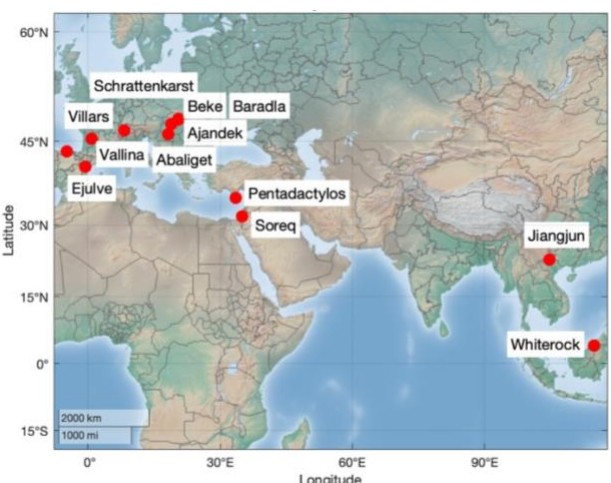

**Figure 3: Map showing locations of cave sites with speleothem temperature records covering Terminations using methods such as degassing corrected $\delta^{13}C$, fluid inclusion hydrogen and oxygen isotopes, clumped isotope thermometry, ratios of Glycerol Dialkyl Glycerol Tetraethers (GDGTs) such as TEX86, noble gas content, and homogenization of fluid inclusions.**

In some mid-latitude regions, speleothem carbon isotopes effectively provide information of the timing of temperature changes at identical temporal resolution to oxygen isotope records (Section 2.3). The initial, temperature-sensitive $\delta^{13}C_{init}$ has been constructed for the NISA record covering TII and the Iberian Range Ejulve cave record covering TIII. The figures with $\delta^{13}C_{corr}$ anomalies ranging from 2 to 4 ‰ suggest an initial temperature decrease at the onset of both TII and TIII, and multiple steps in the warming, capturing millennial features. In TIII, the $\delta^{13}C_{corr}$ record indicates progressive warming coincident with rising insolation.

## 3.4 Records of hydroclimate

### 3.4.1 Monsoon regions

The oxygen isotopic composition of speleothems from the monsoon regions (ISM, EASM, SE Asian and South American) provides a powerful indicator of monsoon variability at orbital timescales (e.g. Cheng et al., 2009; Kathayat et al., 2016; Carolin et al., 2016). Monsoon regions have yielded many speleothems with high amplitude oxygen isotopic anomalies (2-5 ‰) and excellent age control (Supp. Fig. 8). Where the signal amplitude is small, ~2 ‰, for example in the case of South American and SE Asian monsoon records, the impact of ice volume correction on speleothem $\delta^{18}O$ decreases the glacial-interglacial hydrological signal magnitude by ~0.5-1 ‰.



While large amplitude orbital changes in $\delta^{18}O$ precipitation can be confidently inferred in many monsoon systems, relating
the $\delta^{18}O$ variability to specific climatic parameters such as rainfall amount, rainfall seasonality, or moisture source in the
different monsoon regions has been more complicated. Multiple processes may affect the $\delta^{18}O$ of rainfall and dripwater,
including the ocean moisture source isotopic ratio, and the temperature and humidity in the moisture source region, the
degree of moisture recycling over land (re-evaporation) and mixing (Gat, 1996; Maher, 2008), Rayleigh distillation during
progressive rainout between the source and the cave location where the moisture carrying capacity of air parcels is dependent
on temperature (Dansgaard, 1964), seasonality of precipitation (e.g. Chiang et al., 2020) and the intensity of deep convection
(Risi et al., 2008).

In the ISM and EASM systems, the high amplitude oxygen isotopic anomalies suggest that the main driving factor is
changes in the isotopic composition of rainfall, rather than glacial-interglacial cave temperature changes. This has been
confirmed in examples where oxygen isotope ratio or hydrogen isotope ratio of fluid inclusions have been measured directly,
removing the influence of temperature which affects the speleothem calcite $\delta^{18}O$ (Meckler et al., 2015; Wassenburg et al.,
2021). At the eastern extremity of ISM region, fluid inclusions indicate that drip water decreased in $\delta^{18}O$ by 4-5 ‰ over the
deglaciation of TII as a consequence of the hydrological factors discussed above (Wassenburg et al., 2021), and independent
measurements confirm temperatures shifts contributed only ~2 ‰ change in oxygen isotopic variability of the
speleothems. In the large amplitude ISM and EASM termination signals, the ice volume correction is a very small fraction of
the signal.

To first order, the lower speleothem $\delta^{18}O$ in EASM and ISM during interglacials reflect higher temperatures in interglacial
times which increases the moisture carrying capacity of air parcels and resulting increase in rainout upstream of the cave
region and in the cave region itself (Liu et al., 2014), enhanced by changing internal monsoon dynamics (Sinha et al., 2015).
Over the Holocene, both the ISM and the EASM show a lagged response to increasing insolation which is thought to be due
to the presence of ice sheets in the early Holocene (Zhang et al., 2019). Isotope-enabled general circulation simulations
suggest that persistence of wet conditions are thought to reflect the role of land surface and ocean feedbacks in sustaining
regional monsoons (e.g. Dallmeyer et al., 2010; Zhao and Harrison, 2012). Parker et al., 2021 observe a similar lagged
response and persistent wetness across TI. Similarly over TI Pausata et al., (2011) and He et al., (2021) suggest that the more
negative interglacial $\delta^{18}O_{precip}$ in the ISM domain are dominantly driven by the isotope amount effect of enhanced deep
convection. In contrast, the dominant driver of the more negative EASM $\delta^{18}O_{precip}$ is the increased rainout along the moisture
pathway between the Indian Ocean and the EASM region. In models, the coherent pattern of lower interglacial speleothem
$\delta^{18}O$ in the ISM and EASM regions is reflective of continental-scale hydroclimate response marked not by uniform increase
in amount of precipitation, but rather by intensified convection and rainfall over the ISM, and a northward-shifted Westerly
jet leading to increased rainfall in the northern EASM but reduced rainfall in the southern EASM (He et al., 2021). Models





also suggest that periods of weakened Atlantic Meridional Overturning Circulation (AMOC) cause a coherent positive $\delta^{18}O$ anomaly across the ISM and EASM region due to decreased convection over the ISM and decreased rainout between the Indian Ocean and the EASM domain with a southward shifted Westerly jet.


Similar to the ISM region, and the northern EASM region, the SE Asian speleothem records shift to more negative $\delta^{18}O$ values which are representative of wetter conditions and circulation changes during interglacials (Meckler et al., 2012; Parker et al., 2021; Kimbrough et al., 2023). These records are from small islands near the equator. In this setting, the glacial interglacial temperature changes have been found to be in the range of 4-5℃ (Meckler et al., 2015) which given a 2 ‰

change in speleothem $\delta^{18}O$, suggests only about a 1 ‰ change in drip water (and fluid inclusion) $\delta^{18}O$. The depleted values during glacial times may partially result from increased land exposure countering the impact of more positive seawater $\delta^{18}O$ values from glacial enrichment at source (Meckler et al., 2012). Since the Green Cathedral cave records are located ~5°North of the equator while the Abadi cave records are located ~5°South of the equator, the records suggest that during glacial times, the tropical rainbelt was shifted even further below 5°South and/or that there was a significant reduction

in convection over the Indo-Pacific Warm Pool (IPWP) in turn driven by changes in land mass configurations (DiNezio and Tierney, 2013; Di Nezio et al., 2016; Du et al., 2021). During Heinrich Stadial 1, for example, the slowdown of the AMOC resulted in drying over the IPWP (Buckingham et al., 2022). Some modelling results show a reduction in rainfall in the mid-Holocene but this is less than would have been expected in the absence of Indian Ocean feedback. This region may be sensitive to sea surface temperatures (SSTs) in the Indian Ocean and South China Sea regions which are in turn modulated

by Northern Hemisphere winter temperatures (Zhao and Harrison, 2012).

In South America, the moisture at the Diamante cave site is sourced from the tropical Atlantic. The air parcels travel over the Amazon incorporating recycled moisture while travelling westward toward the Andes (Cheng et al., 2013). The west Amazon Diamante cave record shows a smaller amplitude change (~2‰) to increased $\delta^{18}O$ thought to reflect a shift from

relatively wet to moderately dry conditions during the interglacial. The impact of ice volume correction on speleothem $\delta^{18}O$ dampens the glacial anomaly resulting in a decreased glacial-interglacial hydrological signal magnitude by ~0.5-1 ‰. The dominant moisture delivered to the west Amazonian site is thought to be recycled transpiration water from the Amazon basin (25-60% based on different moisture-tagging studies) which is a non-fractionating process, so that the actual rainfall decrease at the west Amazonian site may be even lesser than suggested by the $\delta^{18}O$ change (Cheng et al., 2013).

**3.4.2 Non-monsoon regions**

Hydroclimate is also inferred to be the dominant control on speleothem $\delta^{18}O$ variability in non-monsoon regions in Central Asia, parts of southern European and the Middle East. The oxygen isotope variability of Central Asian records is controlled



by the north-south movement of the Westerly Jet which controls moisture source location with warm periods corresponding
to longer transport pathways and lighter $\delta^{18}O$ values (Supp. Fig. 9) (Liu et al., 2015).

In western Italy, warmer sea surface and air temperatures during interglacials are believed to promote greater rainfall leading to more depleted speleothem $\delta^{18}O$ values reflected by a $\delta^{18}O$ anomaly of ~3 ‰ (Drysdale et al., 2009). Over Terminations, the western Italy Corchia cave record is also affected by changes in the $\delta^{18}O_{seawater}$ of the North Atlantic as well as western
Mediterranean which are the dominant moisture sources. Over TII, the removal of the regional eastern North Atlantic $\delta^{18}O$ variation (from NISA Garth record) provides a pattern of enhanced interglacial rainfall compared to glacial reflected by a $\delta^{18}O$ anomaly of ~3 ‰, but a notable millennial-scale anomaly of 2 ‰ suggesting a transient period of extreme aridity at the onset of the Termination.

The Middle Eastern stalagmite $\delta^{18}O$ records from Soreq, Jerusalem West and Peqiin caves (Supp. Fig. 2) reflect changes in precipitation amount, more depleted $\delta^{18}O$ values suggesting increased precipitation during interglacial times (Bar-Matthews et al., 2003 and references therein). An additional source water composition effect has also been suggested with evaporation in the semi-closed Mediterranean amplifying the melt-water signal (Frumkin et al., 1999).

In eastern North America, the $\delta^{18}O$ record of Buckeye Creek Cave is thought to reflect the annual balance of moisture between Gulf of Mexico or continent sourced heavier $\delta^{18}O_{precip}$ during the summer, and Pacific sourced lighter $\delta^{18}O_{precip}$ during the winter (Supp. Fig. 10). A similar mechanism is thought to play out on longer glacial-interglacial timescales with glacial times of more light $\delta^{18}O_{precip}$ Pacific moisture transport and interglacial timescales of more heavy Gulf of Mexico/continental moisture transport (Hardt et al., 2010; Cheng et al., 2019). The amplitude of the signal is small ~ 1-2 ‰ and the signal is
further dampened by ice volume corrections. The ice volume corrected Buckeye Creek record suggests an earlier and more continuous return to glacial conditions at the end of TIIIA.

## 4 Sequence of climate changes during Terminations as tracked by speleothem records

The diverse aspects of climate recorded by available speleothem records allow us to evaluate if such aspects of the climate
system change simultaneously or in succession across the Termination given the uncertainty in the individual speleothem chronologies. In cases where the ice volume effects have minimal impact on the timing of events (as previously detailed in Section 3), the original records *without* ice volume correction have been plotted in order to preserve the high resolution of the original records which are relevant to discussing the timing and sequence of events over the Terminations.





## 4.1 Termination II

The onset of increased insolation is seen by 137,000 years BP (Fig. 4). The first manifestation of TII in speleothems is the freshening in the North Atlantic from the NISA (La Vallina cave – Garth record) record at 135,676 (+/- 1186) years BP, synchronous with a positive shift in the EASM $\delta^{18}$O by 135,650 (+457/-632) years BP. Such a positive shift in the EASM is conventionally interpreted as the onset of a 'weak monsoon interval' (Cheng et al., 2016b), potentially reflecting a southward shift of the Westerly jet reflected by the Central Asian winter-precipitation driven Tonnelnaya record (Cheng et al., 2016a) and often attributed to reduction in AMOC, consistent with models of TI (He et al., 2021). The sequence confirms with absolute chronology that the first deglacial freshening may trigger synchronous response in the EASM system including on millennial scale, as hypothesised previously. The available ISM records do not resolve this millennial structure. This trend to higher $\delta^{18}$O in the EASM records is interrupted by a centennial scale strengthening of the monsoon at 134,737 years BP that is clearly seen in the Hulu cave record.

An early warming step is detected in the South European NISA record at 135,429 (+/- 1345) years BP within error of the North European Schneckenloch record at 134,500 (+/- 1649) years BP. When comparing records across different systems, the absolute chronological uncertainties suggest that the freshening and initial onset of warming in South Europe may be within error of each other. However, when comparing in a single stalagmite the freshening signal derived from the $\delta^{18}$O proxy and the temperature signal derived from the $\delta^{13}$C and Mg/Ca proxies, (NISA - La Vallina cave - Garth record), there is some confidence that the first freshening preceded the warming step in the NISA record and in fact coincided with a transient temperature minimum which may reflect AMOC reduction. Thereafter, the gradual warming trend in central North America begins at 133,593 (+/- 2346) years BP, and progressive increase in rainfall in West Italy (Corchia cave) following a sharp aridity event at this location. The gradual warming in central North America may reflect rising greenhouse gas concentrations as well as regional lowering of albedo with ice sheet retreat.

The final climate response is the abrupt shift in the East Asian monsoon record, classically interpreted as the onset of strong monsoon at 129,015 (+/- 719) years BP. Within the chronological uncertainties this event is simultaneous in the central Asian and the ISM, as well as the final step of abrupt warming in coastal and northern Europe as recorded in Abaliget, Sieben Hengste, and NISA temperature records. These changes occur toward the end of the freshening signal observed in the NISA record, and according to tuning of marine and NISA speleothem archives, may coincide with the recovery after the final collapse of the Labrador ice dome (Stoll et al., 2022).





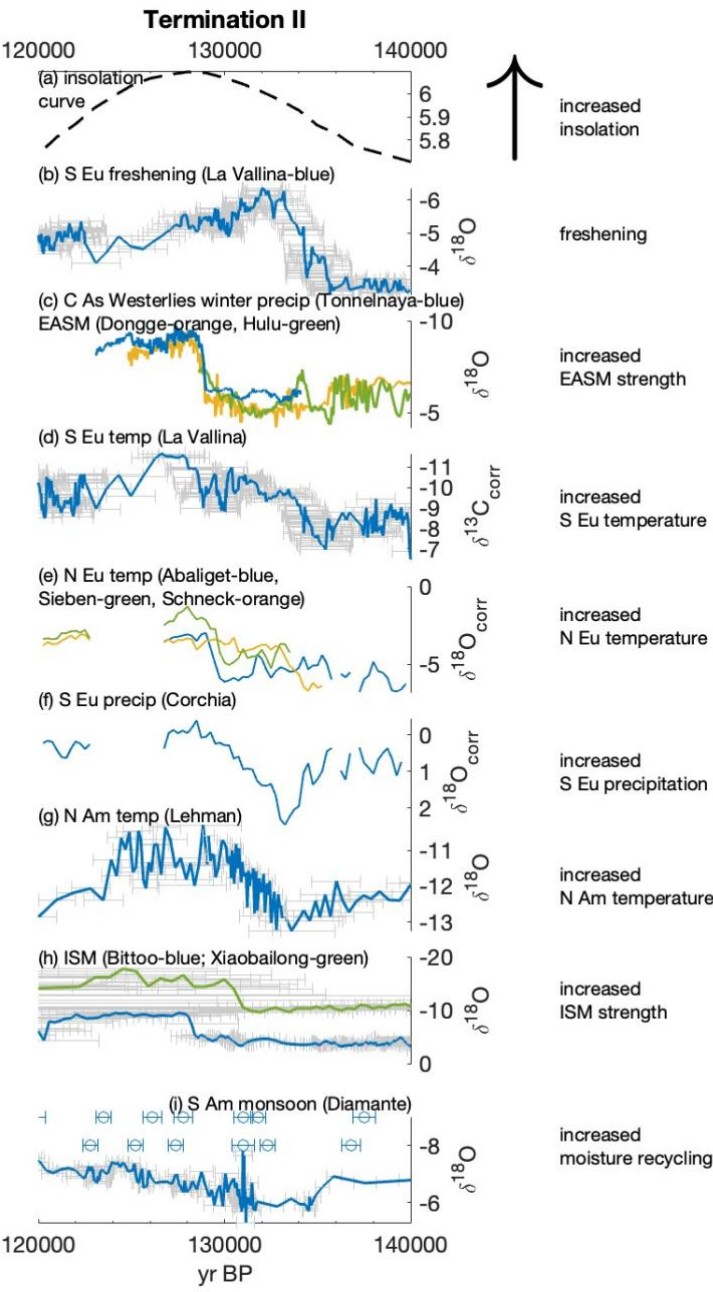

**Figure 4: Ages covering Termination II are plotted against proxy measurements to discuss the sequence of events. Age-depth model uncertainties have also been plotted. $\delta^{18}O_{corr}$ refers to the record which has been previously corrected using the regional ice volume correction (see Section 2.2). $\delta^{13}C_{corr}$ refers to a record which has been previously corrected for degassing (see Section 2.3). Since these corrections involve binning data, no age-depth model uncertainties have been plotted for these records. Insolation curve is the summer half year caloric insolation as provided in (Tzedakis et al., 2017). [precip = precipitation; temp = temperature; N Eu = North Europe; S Eu = South Europe; N Am = North America; C As = Central Asia; ISM = Indian Summer Monsoon; EASM = East Asian Summer Monsoon; SE Asia = Southeast Asia; S Am = South America]**

510





## 4.2 Termination IIIA

The onset of increased insolation is seen by 229,000 years BP (Fig. 5). The first manifestation of TIIIA in available speleothem records is the gradual freshening in the Black Sea region tracked by the Sofular cave speleothem record at 229,085 (+1972/-1512) years BP. A positive $\delta^{18}$O anomaly in the EASM Sanbao record starting at 228,550 years BP coincides with the onset of freshening and may reflect weakened intensity of the ISM and decreased rainout. An early warming step in the European Alps (Spannagel) is seen nearly synchronously with the onset of the Sofular freshening event at 229,099 (+/- 237) years BP. Subsequently the southeastern North American record suggests the onset of a gradual Gulf of Mexico warming (Buckeye Creek) at 227,011 (+/- 2063) years BP. Shortly thereafter (or synchronously given the errors), there is continued intensification of the ISM monsoon centre convection manifest as progressive depletion of EASM $\delta^{18}$O reaching typical interglacial values by 225,783 (+11,284 /-7,146) years BP. There is also an increase in Westerlies driven summer precipitation in the Central Asian Kesang cave at 225,069 years BP.

The SE Asian Abadi cave record (located at 5°S) shows a gradual reduction in monsoon strength between 229,790 (+/- 2407) and 219,200 (+/- 1,122) years BP during the rise in insolation and the Sofular freshening event. In western tropical South America, this time period is marked by an increase in moisture recycling (less upstream rainout) recorded in Diamante Cave. While the Central Asian Kesang and Tonnelnaya cave $\delta^{18}$O cave records are coherent when examined on longer timescales and are similar to EASM records during TII, over TIIIA, the Kesang record does not manifest a positive $\delta^{18}$O shift in response to North Atlantic freshening as seen in the Tonnelnaya and Dongge records from TII. The EASM records show a low amplitude anomaly signal for TIIIA in comparison to TII. One distinct, although still a low amplitude signal, is a period of reduced monsoon strength from 221,189 (+/- 903) to 218,662 (+/- 641) years BP.

The final, strong North Europe warming coincides with the end of the Sofular freshening signal at 217,304 (+/-1972) years BP. The SE Asian monsoon strength (Abadi Cave) begins to increase at this time, together with a parallel reduction in Westerlies driven summer precipitation (Kesang). The EASM experiences a transient positive $\delta^{18}$O anomaly at 218,290 (+/- 1717) years BP and the abrupt recovery at 217,394 (+/- 1,274) years BP is coincident with final North Europe warming.

214,324 (+/- 375) years BP, marks the end of the warming event in the European Alps with decrease in Gulf of Mexico temperatures indicated by the Buckeye Creek cave in North America. The South American Diamante cave does not show a return to pre-Termination conditions by this time.



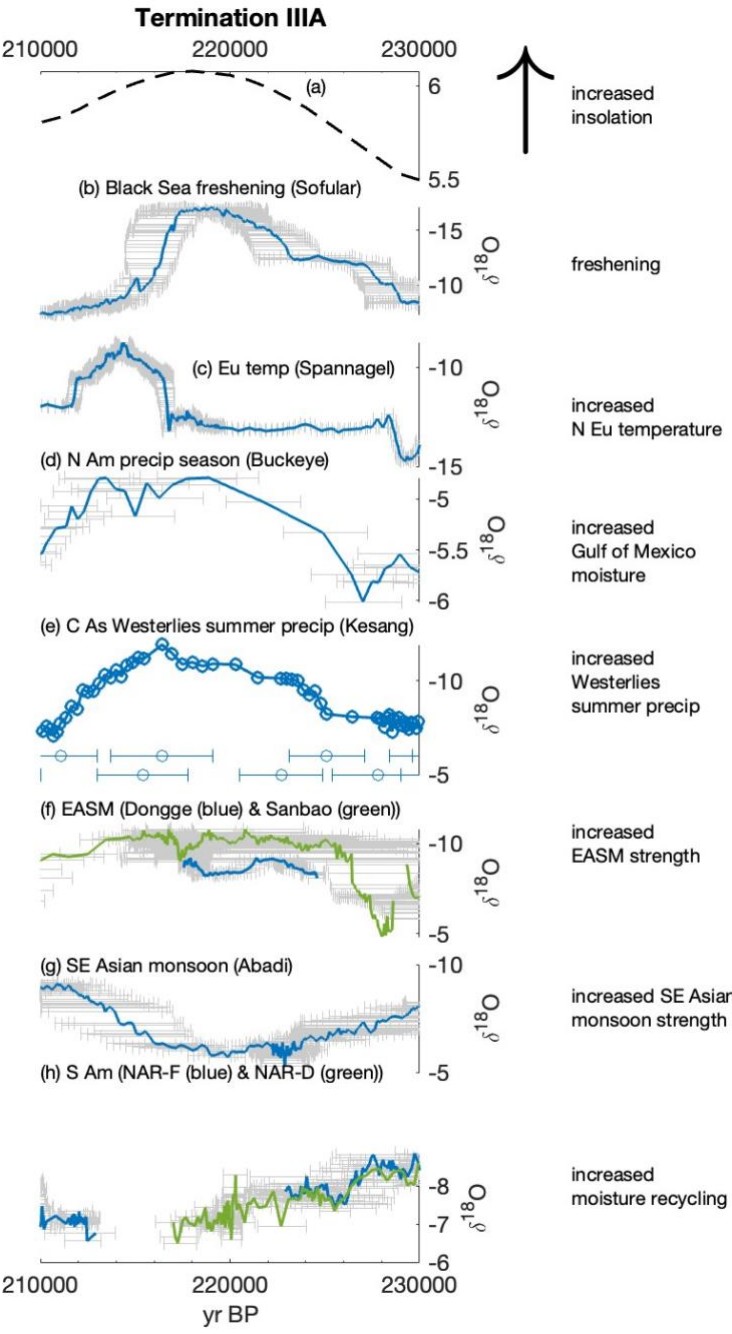

**Figure 5: Ages covering Termination IIIA are plotted against proxy measurements to discuss the sequence of events. Age-depth model uncertainties have also been plotted. Insolation curve is the summer half year caloric insolation as provided in (Tzedakis et al., 2017). [precip = precipitation; temp = temperature; N Eu = North Europe; S Eu = South Europe; N Am = North America; C As = Central Asia; ISM = Indian Summer Monsoon; EASM = East Asian Summer Monsoon; SE Asia = Southeast Asia; S Am = South America]**





### 4.3 Termination III

The onset of increased insolation is seen by 253,000 years BP (Fig. 6). No freshening records are available for TIII. A brief transient positive shift in the EASM $\delta^{18}O$ centred at 252,200 years BP suggests weakened convection intensity and decreased rainout between Indian Ocean and EASM region. This period is also seen as a departure to more positive $\delta^{18}O$ values in the ISM and SE Asian monsoon (Green Cathedral (5°N)) cave records which reflect decreased regional precipitation and changes to circulation. A longer transient positive shift in the EASM record is seen from 248,200 to 243,200 years BP. The

European Ejulve cave record shows gradual warming beginning at 248,545 years BP interrupted by shorter cooling events centred at 243,924 and 241,720 years BP respectively. A prominent warming step is seen in the Alpine Spannagel cave record which only partially covers TIII at 242,446 (+/- 299) years BP.

Similar to the TII Central Asian Tonnelnaya record, and unlike the TIIIA Central Asian Kesang cave record, the Central

Asian Kesang cave record for TIII shows some decrease in Westerlies-driven summer precipitation, though this record has a very low resolution. At the same time, and again conversely to TIIIA, the Southeastern North American record from Buckeye Creek cave suggests a drop in Gulf of Mexico temperatures. However, the signal amplitude for the Buckeye Creek records is less than 1 ‰.

Coinciding with the prominent warming step seen in the European cave records ~ 242,000 years, there is an increase in EASM monsoon strength at 243,200 (+/- 496) years BP. The ISM and SE Asian monsoon strength also change around the same time period but have large chronological errors which precludes further discussion on their place in the sequence of events. Similarly, the change to warmer Gulf of Mexico and linked precipitation season in the Buckeye Creek cave, and Westerlies season in the Kesang cave, all lie within errors of the monsoon change in the EASM region and the temperature

change in the Alps.



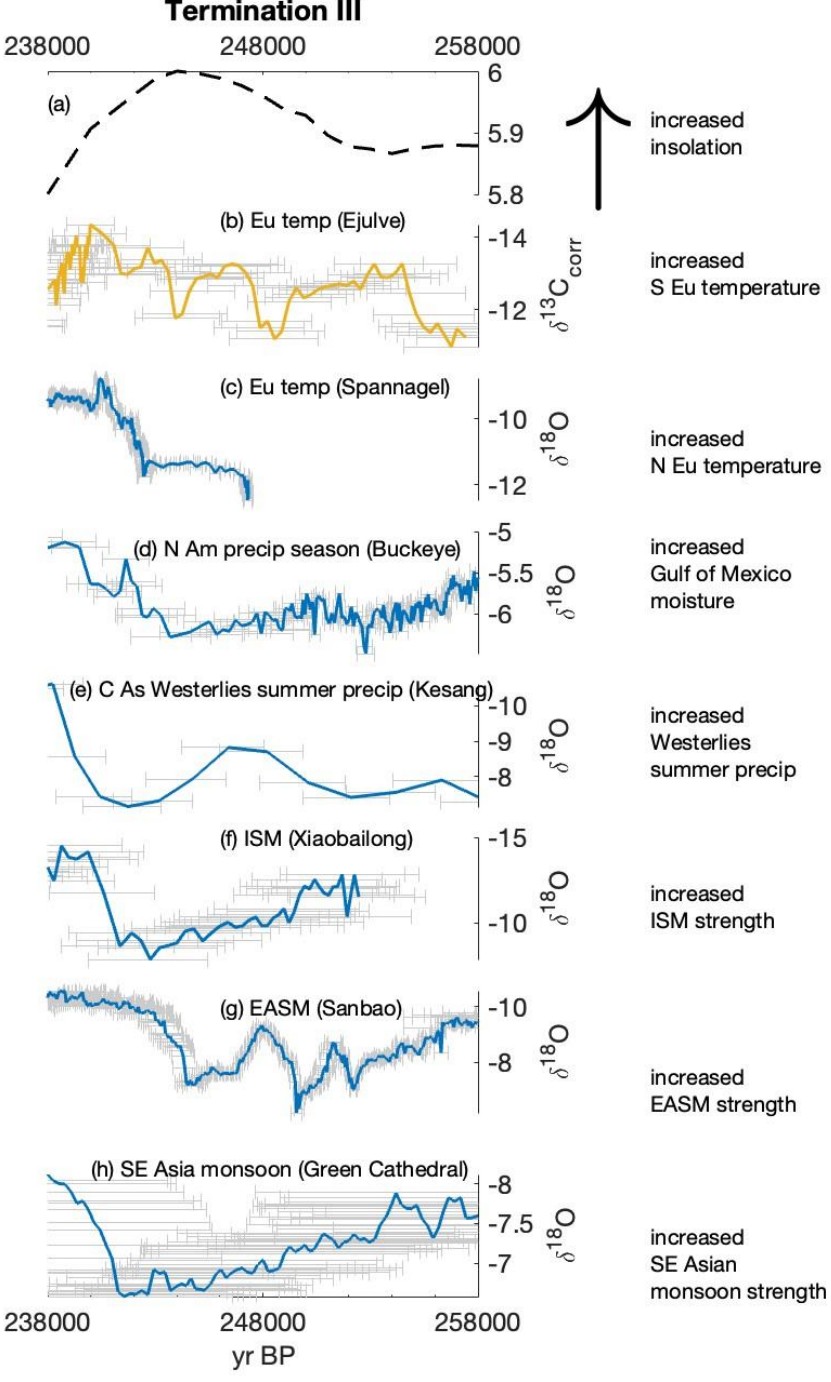

**Figure 6: Ages covering Termination III are plotted against proxy measurements to discuss the sequence of events. Age-depth model uncertainties have also been plotted. Insolation curve is the summer half year caloric insolation as provided in (Tzedakis et al., 2017). [precip = precipitation; temp = temperature; N Eu = North Europe; S Eu = South Europe; N Am = North America; C As = Central Asia; ISM = Indian Summer Monsoon; EASM = East Asian Summer Monsoon; SE Asia = Southeast Asia; S Am = South America]**



### 4.4 TIV and TV

Speleothem records of TIV and TV are only available from the EASM region (Sanbao cave) and the SE Asian monsoon region (Abadi and Green Cathedral caves respectively). No freshening records are available for these Terminations. Similar to TII and TIII, the EASM records indicate weak monsoon periods at the onset of insolation increase. Both EASM and SE Asian monsoon records recover around the same time around peak insolation.

## 5 Discussion

### 5.1 Similarities and differences among Terminations within the same regions

Records of different Terminations from different speleothems belonging to the same cave and region are largely similar in direction of signal and magnitude (with few exceptions), but can vary significantly in internal structure (including the rate of change).


Records of freshening from NISA for TII and Sofular cave for TIIIA are markedly different in magnitude (Supp. Fig. 6). This is a manifestation of the different sources of moisture for the two records rather than a reflection of the differences between Terminations. Freshening of the North Atlantic is thought to have altered the source composition by ~2‰ (Stoll et al., 2022). Inflow of isotopically depleted Caspian Sea waters resulting from Eurasian ice sheet melt, diversion of rivers,

higher river runoff coefficients and reduced evaporation to the Black Sea during Terminations is thought to be the source of the depleted $\delta^{18}O$ values of the Sofular cave record (Badertscher et al., 2011 and references therein). Since there is no corresponding NISA record for TIIIA, it is difficult to gauge the response time of the Caspian Sea to deglaciations, however the sequence of events surrounding the warming in North Europe, and recovery of the EASM, that are present in both TII and TIIIA records, suggest that the Caspian Sea likely reacted rapidly i.e. within ~2000 years age error of the Sofular

speleothem record.

European temperature records are available from Abaliget and NISA for TII, Spannagel for TIIIA, and Ejulve and partially Spannagel for TIII (Supp. Fig. 7). Once the ice volume correction from the NISA record has been applied to the Abaliget record, the structure of the record is similar to the $\delta^{13}C$-based temperature record from NISA (Fig. 4). Currently, no similar

North Atlantic specific correction can be made for the Spannagel $\delta^{18}O$ record from TIIIA and TIII or Ejulve in TIII; in the measured records, times of meltwater arrival in the North Atlantic could mask early warming steps during the times of apparently stable $\delta^{18}O$ between 228 and 215 ka in TIIIA and 242 to 247 ka in TIII. For TIII the Ejulve cave $\delta^{13}C$-based temperature record provides this information instead with a two step temperature increase at 248,545 and 242,446 years BP





respectively. While these records provide excellent information of the timing of temperature increase, they can provide only
an estimate of change in actual temperature. For this we must look to other speleothem records such as fluid inclusions
instead. The Hungarian cave fluid inclusion records (see subsection 3.3) are roughly from the same region and elevation as
the Abaliget record at (~450 and 200 m asl respectively) and suggest a temperature change of 7.4℃. The Schrattenkarst fluid
inclusion record for TII is available from a similar region and elevation (~2000 m) as the Spannagel cave record and
suggests a temperature change of 4.8℃. Only the Hungarian cave fluid inclusion record is available for TIII and suggests
very limited temperature variation at least in this region and at this elevation. This is unlike what is suggested by the
Spannagel cave $\delta^{18}$O record.

The EASM records from Sanbao and Dongge together cover all Terminations with Sanbao records particularly available for
all Terminations. Although from the same region, the records are from different stalagmites with different sampling
resolutions and age uncertainties, nevertheless, they provide an opportunity to examine why the Terminations may have
different structures in the same region (Supp. Fig. 8) (Cheng et al., 2009). TII shows the most abrupt change in $\delta^{18}$O and lasts
for ~6,985 years from the initial weak monsoon event to monsoon recovery. TIV is similar to TII in structure and lasts for
~7,136 years. The peak weak monsoon event for TV seems to last only 2,223 years. TII, TIIIA and TIII show prominent
millennial event structures during Terminations.


The SE Asian monsoon records from Green Cathedral and Abadi caves are another group of records from the same monsoon
region that can be compared across different Terminations. The records from TIIIA, TIII and TV are similar to each other
showing gentle weakening and recovery signals across thousands of years. The TIV Abadi cave record, in contrast, shows a
sharp recovery lasting only a few 100 years.


**5.2 Similarities and differences in the sequence and duration of events during Terminations**

The detailed analysis of available speleothem records covering Terminations in this study allow us to provide a more
structured interpretation of the sequence of events (Fig. 7). The 3 Terminations, namely, TII, TIIIA and TIII, show a
common sequence among some events. In all these Terminations, there are two groups of events which are largely out of age
uncertainty of the other. The first group of global sequence of climatic events is represented by the onset of insolation
increase, onset of freshening, weakening of the Asian monsoon, initial increase in temperatures in Europe and North
America. This is followed by a second group of global sequence of climatic events of the end of freshening, near peak
insolation, including monsoon recovery and final step of temperature increase in Europe and North America. Within the
groups, the individual climatic events can have overlapping uncertainties. Where events within a group have no overlap, it is
possible to identify the sequence of events with finer detail. These records provide promising targets for understanding



climate dynamics and for modelling studies. A notable example of such a target includes the onset and time period of warming versus freshening in the North Atlantic region from the NISA $\delta^{18}$O and $\delta^{13}$C records available for TII.

The sequence of events, if bookended by weakening and recovery of the monsoon, seem to cover ~ 8000 years reflected with onset of freshening and recovery of monsoon. However, because the different Terminations feature speleothems from different regions, it is currently not possible to precisely evaluate variations in the duration of the different Terminations or processes that contribute to such variations.

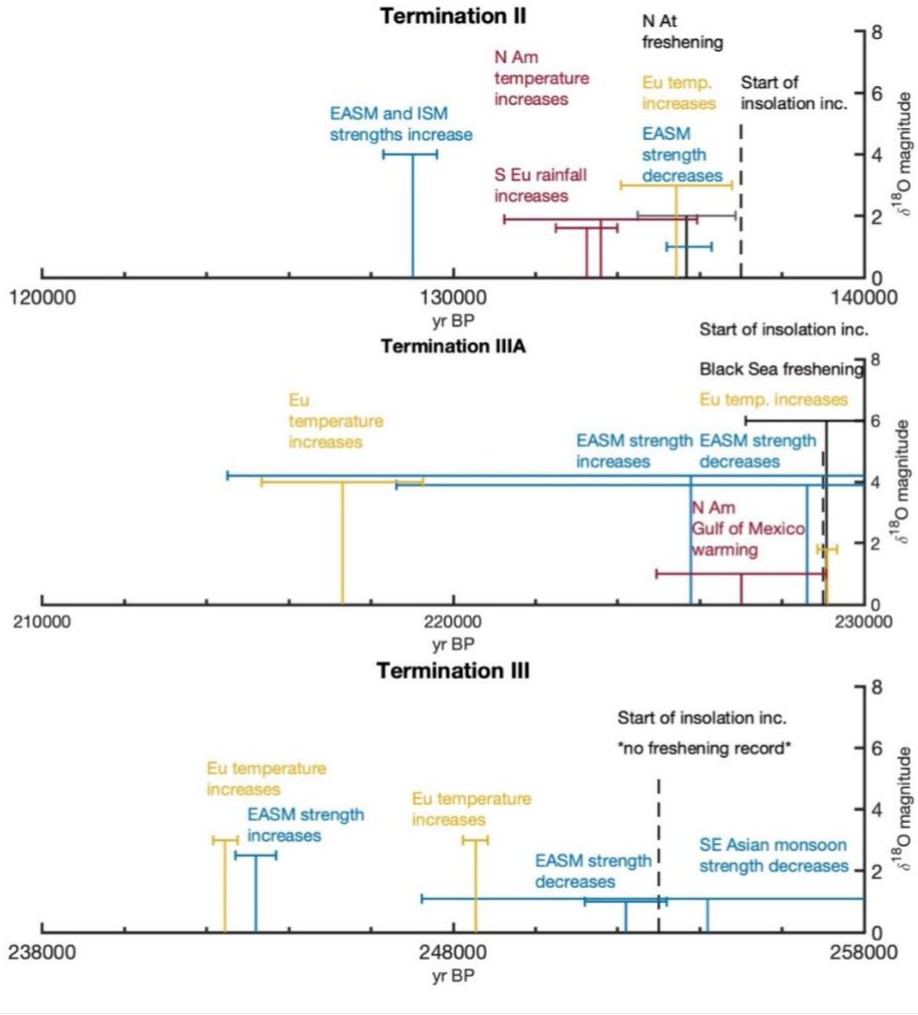


**Figure 7: Sequence of global climatic events over Terminations. Ages and chronological uncertainties are represented on the X-axes. Amplitude of oxygen isotope changes that reflect the climatic events in speleothem records are plotted on the Y axes. The dashed line shows the start of insolation increase. [precip = precipitation; temp = temperature; N Eu = North Europe; S Eu = South Europe; N Am = North America; C As = Central Asia; ISM = Indian Summer Monsoon; EASM = East Asian Summer Monsoon; SE Asia = Southeast Asia; S Am = South America]**



### 5.3 Potential use of speleothem for tuning other records - targets

Several approaches have been employed to transfer speleothem chronologies to marine sediment records by tuning marine proxies to regionally relevant speleothem $\delta^{18}O$ records. Tzedakis et al., (2018) match Iberian margin pollen records with Corchia (Italy) speleothem $\delta^{18}O$, assuming a common rainfall amount signal dominates both records. Grant et al., (2012) tune planktonic $\delta^{18}O$ from the eastern Mediterranean region to speleothem $\delta^{18}O$ records from Israel, assuming the signal of Mediterranean seawater $\delta^{18}O$ dominates both records. In the North Atlantic region, the NISA speleothem records of eastern north Atlantic surface ocean $\delta^{18}O$, have been used to tune Iberian margin marine records of $\delta^{18}O_{seawater}$ derived from paired Mg/Ca and oxygen isotope measurements of planktic foraminifera (Stoll et al., 2022). Caballero-Gill et al., (2012) tune planktonic $\delta^{18}O$ from the South China Sea (SCS) to the East Asian Summer monsoon $\delta^{18}O$ assuming that fresher conditions in the SCS correspond to monsoon regimes reflected in lower speleothem $\delta^{18}O$. This chronology has been applied to evaluate the synchronicity of changes in precipitation isotopes across a the East Asian Monsoon domain, by comparing the speleothem records with leaf wax hydrogen isotope records from same core, since the leaf waxes integrate the signal from the Pearl River catchment area (Thomas et al., 2014). This exercise explores whether monsoon changes result from direct insolation forcing or additionally from SST and ice volume changes, a comparison not possible when marine record chronology of ice volume is derived from tuning to a specific insolation target. In other settings, closely located records of leaf wax hydrogen isotopes and speleothem $\delta^{18}O$, which can be assumed to receive similar isotopical signals in precipitation, may provide further opportunities to tune marine sediment cores. Further leaf wax $\delta^2H$ are available in SE Asian (Sumatra), EASM, and South European (Mediterranean) regions. Finally, emerging records of fluid inclusion thermometry covering Terminations also provide an opportunity to improve chronologies for nearby marine records by tuning the SST indicators from marine cores to speleothem temperature records.

In some cases, speleothem chronology has been applied to more distant archives inferred to feature common forcing. (Grant et al., 2014) tune millennial dust peaks in sediment records from the Red Sea with millennial 'weak monsoon' positive anomalies in speleothem $\delta^{18}O$ records from the Asian monsoon region, inferring synchronous phases of aridity, in order to apply absolute chronology to the Red Sea sea level record. Sinnl et al., (2023) have tuned polar ice core chronology over the last glacial to speleothem chronology using common radionuclide production signals. The potential for precise global speleothem chronologies to tune Heinrich events in both Greenland and Antarctic ice cores on the basis of major atmospheric circulation reorganisations has been proposed (Dong et al., 2022). Over orbital timescales, EASM $\delta^{18}O$ variations have been used to validate the ice core age model, assuming EASM $\delta^{18}O$ correlated with tropical rainfall and NH tropical wetland extent, which would be reflected in the ice core methane concentrations (Suwa and Bender, 2008). More widespread, well dated speleothem records of Terminations beyond TII are required, and development of marine proxy records with signals common to the speleothem records, will continue to improve the chronology of Terminations.






### 5.4 Suitable modelling outputs to understand climate dynamics

The speleothem record of the sequence of climate events over Terminations can be used to evaluate climate model skill of teleconnections and feedbacks (Cheng et al., 2009; Denton et al., 2010; Obase et al., 2021; He et al., 2021). However, to date, few transient simulations covering Terminations exist (e.g. Liu et al., 2009; Ivanovic et al., 2016; Obase et al., 2021; Menviel et al., 2019; Caley et al., 2014) and most of them extend back only to TI. Model outputs can vary based on initial conditions, for example, the manifestation of millennial events surrounding Terminations in model simulations varies based on the sensitivity of different models to different background conditions and the treatment of deglacial meltwater input to the ocean. Some models are also better at simulating certain regions compared to other models making a multi-model comparison more valuable (Bong et al., 2024).


Isotope-enabled model simulations allow direct comparison of rainwater $\delta^{18}O$ simulated by models to calculated rainwater $\delta^{18}O$ from speleothems $\delta^{18}O$, and are run by ECHAM5-wiso (e.g. Comas-Bru et al., 2019), iLOVECLIM (e.g. Caley et al., 2014), iTRACE (e.g. He et al., 2021), iCESM (e.g. Sinha et al., 2024) and GISS (e.g. Tierney et al., 2011). For example, the isotope enabled models have confirmed that synchronous isotope anomalies in the Asian monsoon region reflect a coherent response to large scale reorganisation of atmospheric circulation, even when there are disparate regional responses to precipitation amount (Caley et al., 2014; He et al., 2021). Further, if models confirm, for example, isotope anomalies in the Asian monsoon region are synchronous with expansion and contraction of Northern Hemisphere tropical wetlands, this would provide strengthened support for tuning ice core methane records to the EASM speleothem chronology. By clarifying the processes responsible for the isotopic signal in speleothems, isotope models foster more robust comparison between speleothem records and suitable climate variables simulated by the larger group of non-isotope enabled climate models.

Since there is a protocol for isotope-enabled model comparison for TII (Menviel et al., 2019) and investigation by the current review paper has shown that there are a number of speleothem proxy records from around the globe covering this Termination, TII can be a useful future target to understand climate dynamics using data-model comparison approaches.

**6 Further work**

Based on this study, we can now provide strategic future research directions for speleothem research covering Terminations:

- The NISA records provide the best data on freshening events in the North Atlantic and temperature changes in Southern Europe. Since multiple proxies from the same speleothems provide information of both temperature
change and freshening, they provide critical chronological control on these two events. Further, since at least some




speleothems from this region allow for confocal layer counting, it greatly increases our confidence in the duration of events in this critical source region.

- The Central Asian Tonnel'naya record provides the most clear manifestation of changes to the Westerlies during TII which are also relevant to the coupling between (or forcing of) the North Atlantic and EASM regions. It would be useful to have further records of Westerlies track changes across other Terminations.

- The EASM region with its rich record of well-dated highly resolved speleothems provides the best opportunity to understand the difference in the internal structure of Termination events in speleothem $\delta^{18}O$ records across all examined Terminations. Comparison of different insolation metrics and isotope-enabled modelling studies possibly provide future directions to understanding the variable internal structures in turn providing a hint at the differences in the forcings of different Terminations.

- Current available studies suggest that South American speleothems provide our best opportunity of finding well-dated records across Terminations from the Southern Hemisphere. At the moment, the South American records from the SISAL database cover only TII and TIIIA.

- The North American Lehman caves record provides the best data for temperature changes in this region. However, the Lehman caves record only provides information for TII. The Devils Hole cave records obtained from sub-aqueous speleothems are more likely to cover the older Terminations. However, these sub-aqueous records initially generated debate on their chronological control which have since been resolved (see Moseley et al., 2015 and references therein). However, there continues to be debate on travel time of the climatic signal at the surface to the site of the sub-aqueous record. Further karst modelling studies can provide this information.

- SE Asian speleothem records not only provide critical information of changes over the equator, but possibly also have some of the best understood fluid inclusion records. These are particularly vital since they can be an important tuning target with marine records given their location in the Indo-Pacific Warm Pool.

- Many speleothem records that have been plotted in this study, were not initially measured targeting Terminations. Further, speleothem uranium-thorium dating capabilities and techniques such as confocal microscopy can provide further age control in some of the older records. In such cases, it may be worth undertaking a concentrated effort to improve the parts of the records that cover Terminations, particularly targeting for example, NISA, Northern Europe and EASM records that are within uncertainty of each other .

- Where multiple records are available for a given region, we can evaluate the scale of local to regional coherency in the proxy interpretation. Regional speleothem age models that encompass the strengths and uncertainties from uranium-thorium dates from multiple overlapping speleothems from the same and nearby caves can be attempted.

- More advanced time series methods such as change point analysis can be used which consider full age uncertainty ensembles.





- Records from other speleothem proxies such as trace element and $\delta^{234}$U measurements are increasing in number and provide information of regional hydroclimate and vegetation changes providing a more complete picture of changing climate and environmental conditions in response to Termination events.
- Lastly, we can examine the expression of millennial and more abrupt events during Terminations.

**Competing Interests**

The contact author has declared that none of the authors has any competing interests.

**Acknowledgements.**

This study includes data compiled by SISAL (Speleothem Isotopes Synthesis and AnaLysis), a Working Group of the Past Global Changes (PAGES) project. PAGES received support from the Swiss Academy of Sciences and the Chinese Academy of Sciences. The authors of this study are grateful to the SISAL Working Group for compiling the speleothem database.

**Financial support.**

Nikita Kaushal was funded by the Swiss Government Excellence Scholarship. Carlos Perez-Mejias was funded by National Nature Science Foundation of China (NSFC) grant 42050410317.

**Data Availability**

Data used in this study are available in the SISALv3 database (Kaushal et al., 2024a).



Adkins, J. F., McIntyre, K., and Schrag, D. P.: The Salinity, Temperature, and δ18O of the Glacial Deep Ocean, Science, 298, 1769–1773, https://doi.org/10.1126/science.1076252, 2002.

Affolter, S., Steinmann, P., Aemisegger, F., Purtschert, R., and Leuenberger, M.: Origin and percolation times of Milandre Cave drip water determined by tritium time series and beryllium-7 data from Switzerland, J. Environ. Radioact., 222, 106346, https://doi.org/10.1016/j.jenvrad.2020.106346, 2020.

Alley, R. B. and Cuffey, K. M.: 9. Oxygen- and Hydrogen-Isotopic Ratios of Water in Precipitation: Beyond Paleothermometry, in: 9. Oxygen- and Hydrogen-Isotopic Ratios of Water in Precipitation: Beyond Paleothermometry, De 785 Gruyter, 527–554, https://doi.org/10.1515/9781501508745-012, 2018.

Amirnezhad-Mozhdehi, S. and Comas-Bru, L.: MATLAB scripts to produce OxCal chronologies for SISAL database (scripts V1), , https://doi.org/10.5281/zenodo.3586280, 2019.

Atsawawaranunt, K., Harrison, S., and Comas Bru, L.: SISAL (Speleothem Isotopes Synthesis and AnaLysis Working Group) database Version 1.0, https://doi.org/10.17864/1947.147, 2018a.

Atsawawaranunt, K., Comas-Bru, L., Amirnezhad Mozhdehi, S., Deininger, M., Harrison, S. P., Baker, A., Boyd, M., Kaushal, N., Ahmad, S. M., Ait Brahim, Y., Arienzo, M., Bajo, P., Braun, K., Burstyn, Y., Chawchai, S., Duan, W., Hatvani, I. G., Hu, J., Kern, Z., Labuhn, I., Lachniet, M., Lechleitner, F. A., Lorrey, A., Pérez-Mejías, C., Pickering, R., Scroxton, N., and SISAL Working Group Members: The SISAL database: a global resource to document oxygen and carbon isotope records from speleothems, Earth Syst. Sci. Data, 10, 1687–1713, https://doi.org/10.5194/essd-10-1687-2018, 2018b.

Atsawawaranunt, K., Harrison, S., and Comas-Bru, L.: SISAL (Speleothem Isotopes Synthesis and AnaLysis Working Group) database version 1b, https://doi.org/10.17864/1947.189, 2019.

Badertscher, S., Fleitmann, D., Cheng, H., Edwards, R. L., Göktürk, O. M., Zumbühl, A., Leuenberger, M., and Tüysüz, O.: Pleistocene water intrusions from the Mediterranean and Caspian seas into the Black Sea, Nat. Geosci., 4, 236–239, https://doi.org/10.1038/ngeo1106, 2011.

Barker, S. and Knorr, G.: Millennial scale feedbacks determine the shape and rapidity of glacial termination, Nat. Commun., 12, 2273, https://doi.org/10.1038/s41467-021-22388-6, 2021.

Bar-Matthews, M., Ayalon, A., Gilmour, M., Matthews, A., and Hawkesworth, C. J.: Sea–land oxygen isotopic relationships from planktonic foraminifera and speleothems in the Eastern Mediterranean region and their implication for paleorainfall during interglacial intervals, Geochim. Cosmochim. Acta, 67, 3181–3199, https://doi.org/10.1016/S0016-7037(02)01031-1, 805 2003.

Bengtson, S. A., Menviel, L. C., Meissner, K. J., Missiaen, L., Peterson, C. D., Lisiecki, L. E., and Joos, F.: Lower oceanic $\delta^{13}C$ during the last interglacial period compared to the Holocene, Clim. Past, 17, 507–528, https://doi.org/10.5194/cp-17-507-2021, 2021.

Bong, H., Cauquoin, A., Okazaki, A., Chang, E.-C., Werner, M., Wei, Z., Yeo, N., and Yoshimura, K.: Process-Based 810 Intercomparison of Water Isotope-Enabled Models and Reanalysis Nudging Effects, J. Geophys. Res. Atmospheres, 129, e2023JD038719, https://doi.org/10.1029/2023JD038719, 2024.

Breitenbach, S. F. M., Adkins, J. F., Meyer, H., Marwan, N., Kumar, K. K., and Haug, G. H.: Strong influence of water vapor source dynamics on stable isotopes in precipitation observed in Southern Meghalaya, NE India, Earth Planet. Sci. Lett., 292, 212–220, https://doi.org/10.1016/j.epsl.2010.01.038, 2010.



Brook, E. J. and Buizert, C.: Antarctic and global climate history viewed from ice cores, Nature, 558, 200–208, https://doi.org/10.1038/s41586-018-0172-5, 2018.

Buckingham, F. L., Carolin, S. A., Partin, J. W., Adkins, J. F., Cobb, K. M., Day, C. C., Ding, Q., He, C., Liu, Z., Otto-Bliesner, B., Roberts, W. H. G., Lejau, S., and Malang, J.: Termination 1 Millennial-Scale Rainfall Events Over the Sunda Shelf, Geophys. Res. Lett., 49, e2021GL096937, https://doi.org/10.1029/2021GL096937, 2022.

Caballero-Gill, R. P., Clemens, S. C., and Prell, W. L.: Direct correlation of Chinese speleothem δ18O and South China Sea planktonic δ18O: Transferring a speleothem chronology to the benthic marine chronology, Paleoceanography, 27, https://doi.org/10.1029/2011PA002268, 2012.

Cai, Y., Fung, I. Y., Edwards, R. L., An, Z., Cheng, H., Lee, J.-E., Tan, L., Shen, C.-C., Wang, X., Day, J. A., Zhou, W., Kelly, M. J., and Chiang, J. C. H.: Variability of stalagmite-inferred Indian monsoon precipitation over the past 252,000 y,
Proc. Natl. Acad. Sci., 112, 2954–2959, https://doi.org/10.1073/pnas.1424035112, 2015.

Caillon, N., Severinghaus, J. P., Jouzel, J., Barnola, J.-M., Kang, J., and Lipenkov, V. Y.: Timing of Atmospheric CO2 and Antarctic Temperature Changes Across Termination III, Science, 299, 1728–1731, https://doi.org/10.1126/science.1078758, 2003.

Caley, T., Roche, D. M., Waelbroeck, C., and Michel, E.: Constraining the Last Glacial Maximum climate by data-model
(<i>i</i>LOVECLIM) comparison using oxygen stable isotopes, https://doi.org/10.5194/cpd-10-105-2014, 10 January 2014.

Capron, E., Rovere, A., Austermann, J., Axford, Y., Barlow, N. L. M., Carlson, A. E., de Vernal, A., Dutton, A., Kopp, R. E., McManus, J. F., Menviel, L., Otto-Bliesner, B. L., Robinson, A., Shakun, J. D., Tzedakis, P. C., and Wolff, E. W.: Challenges and research priorities to understand interactions between climate, ice sheets and global mean sea level during
past interglacials, Quat. Sci. Rev., 219, 308–311, https://doi.org/10.1016/j.quascirev.2019.06.030, 2019.

Carlson, A. E. and Winsor, K.: Northern Hemisphere ice-sheet responses to past climate warming, Nat. Geosci., 5, 607–613, https://doi.org/10.1038/ngeo1528, 2012.

Carolin, S. A., Cobb, K. M., Lynch-Stieglitz, J., Moerman, J. W., Partin, J. W., Lejau, S., Malang, J., Clark, B., Tuen, A. A., and Adkins, J. F.: Northern Borneo stalagmite records reveal West Pacific hydroclimate across MIS 5 and 6, Earth Planet.
Sci. Lett., 439, 182–193, https://doi.org/10.1016/j.epsl.2016.01.028, 2016.

Cheng, H., Edwards, R. L., Wang, Y., Kong, X., Ming, Y., Kelly, M. J., Wang, X., Gallup, C. D., and Liu, W.: A penultimate glacial monsoon record from Hulu Cave and two-phase glacial terminations, Geology, 34, 217–220, https://doi.org/10.1130/G22289.1, 2006.

Cheng, H., Edwards, R. L., Broecker, W. S., Denton, G. H., Kong, X., Wang, Y., Zhang, R., and Wang, X.: Ice Age
Terminations, Science, 326, 248–252, https://doi.org/10.1126/science.1177840, 2009.

Cheng, H., Sinha, A., Cruz, F. W., Wang, X., Edwards, R. L., d'Horta, F. M., Ribas, C. C., Vuille, M., Stott, L. D., and Auler, A. S.: Climate change patterns in Amazonia and biodiversity, Nat. Commun., 4, 1411, https://doi.org/10.1038/ncomms2415, 2013.

Cheng, H., Spötl, C., Breitenbach, S. F. M., Sinha, A., Wassenburg, J. A., Jochum, K. P., Scholz, D., Li, X., Yi, L., Peng, Y.,
Lv, Y., Zhang, P., Votintseva, A., Loginov, V., Ning, Y., Kathayat, G., and Edwards, R. L.: Climate variations of Central Asia on orbital to millennial timescales, Sci. Rep., 6, 36975, https://doi.org/10.1038/srep36975, 2016a.





Cheng, H., Edwards, R. L., Sinha, A., Spötl, C., Yi, L., Chen, S., Kelly, M., Kathayat, G., Wang, X., Li, X., Kong, X., Wang, Y., Ning, Y., and Zhang, H.: The Asian monsoon over the past 640,000 years and ice age terminations, Nature, 534, 640–646, https://doi.org/10.1038/nature18591, 2016b.

Cheng, H., Springer, G. S., Sinha, A., Hardt, B. F., Yi, L., Li, H., Tian, Y., Li, X., Rowe, H. D., Kathayat, G., Ning, Y., and Edwards, R. L.: Eastern North American climate in phase with fall insolation throughout the last three glacial-interglacial cycles, Earth Planet. Sci. Lett., 522, 125–134, https://doi.org/10.1016/j.epsl.2019.06.029, 2019.

Chiang, J. C. H., Herman, M. J., Yoshimura, K., and Fung, I. Y.: Enriched East Asian oxygen isotope of precipitation indicates reduced summer seasonality in regional climate and westerlies, Proc. Natl. Acad. Sci., 117, 14745–14750, 860 https://doi.org/10.1073/pnas.1922602117, 2020.

Comas-Bru, L., Harrison, S. P., Werner, M., Rehfeld, K., Scroxton, N., Veiga-Pires, C., and SISAL working group members: Evaluating model outputs using integrated global speleothem records of climate change since the last glacial, Clim. Past, 15, 1557–1579, https://doi.org/10.5194/cp-15-1557-2019, 2019.

Comas-Bru, L., Atsawawaranunt, K., Harrison, S., and members, S. working group: SISAL (Speleothem Isotopes Synthesis 865 and AnaLysis Working Group) database version 2.0, https://doi.org/10.17864/1947.256, 2020a.

Comas-Bru, L., Rehfeld, K., Roesch, C., Amirnezhad-Mozhdehi, S., Harrison, S. P., Atsawawaranunt, K., Ahmad, S. M., Brahim, Y. A., Baker, A., Bosomworth, M., Breitenbach, S. F. M., Burstyn, Y., Columbu, A., Deininger, M., Demény, A., Dixon, B., Fohlmeister, J., Hatvani, I. G., Hu, J., Kaushal, N., Kern, Z., Labuhn, I., Lechleitner, F. A., Lorrey, A., Martrat, B., Novello, V. F., Oster, J., Pérez-Mejías, C., Scholz, D., Scroxton, N., Sinha, N., Ward, B. M., Warken, S., Zhang, H., and 870 SISAL Working Group members: SISALv2: a comprehensive speleothem isotope database with multiple age–depth models, Earth Syst. Sci. Data, 12, 2579–2606, https://doi.org/10.5194/essd-12-2579-2020, 2020b.

Dallmeyer, A., Claussen, M., and Otto, J.: Contribution of oceanic and vegetation feedbacks to Holocene climate change in monsoonal Asia, Clim. Past, 6, 195–218, https://doi.org/10.5194/cp-6-195-2010, 2010.

Dansgaard, W.: The O18-abundance in fresh water, Geochim. Cosmochim. Acta, 6, 241–260, https://doi.org/10.1016/0016-875 7037(54)90003-4, 1954.

Dansgaard, W.: Stable isotopes in precipitation, Tellus, 16, 436–468, https://doi.org/10.3402/tellusa.v16i4.8993, 1964.

Demény, A., Rinyu, L., Kern, Z., Hatvani, I. G., Czuppon, G., Surányi, G., Leél-Őssy, S., Shen, C.-C., and Koltai, G.: Paleotemperature reconstructions using speleothem fluid inclusion analyses from Hungary, Chem. Geol., 563, 120051, https://doi.org/10.1016/j.chemgeo.2020.120051, 2021.

Denton, G. H., Alley, R. B., Comer, G. C., and Broecker, W. S.: The role of seasonality in abrupt climate change, Quat. Sci. Rev., 24, 1159–1182, https://doi.org/10.1016/j.quascirev.2004.12.002, 2005.

Denton, G. H., Anderson, R. F., Toggweiler, J. R., Edwards, R. L., Schaefer, J. M., and Putnam, A. E.: The Last Glacial Termination, Science, 328, 1652–1656, https://doi.org/10.1126/science.1184119, 2010.

Di Nezio, P. N., Timmermann, A., Tierney, J. E., Jin, F.-F., Otto-Bliesner, B., Rosenbloom, N., Mapes, B., Neale, R., 885 Ivanovic, R. F., and Montenegro, A.: The climate response of the Indo-Pacific warm pool to glacial sea level, Paleoceanography, 31, 866–894, https://doi.org/10.1002/2015PA002890, 2016.

DiNezio, P. N. and Tierney, J. E.: The effect of sea level on glacial Indo-Pacific climate, Nat. Geosci., 6, 485–491, https://doi.org/10.1038/ngeo1823, 2013.



Dong, X., Kathayat, G., Rasmussen, S. O., Svensson, A., Severinghaus, J. P., Li, H., Sinha, A., Xu, Y., Zhang, H., Shi, Z.,
Cai, Y., Pérez-Mejías, C., Baker, J., Zhao, J., Spötl, C., Columbu, A., Ning, Y., Stríkis, N. M., Chen, S., Wang, X., Gupta, A.
K., Dutt, S., Zhang, F., Cruz, F. W., An, Z., Lawrence Edwards, R., and Cheng, H.: Coupled atmosphere-ice-ocean dynamics
during Heinrich Stadial 2, Nat. Commun., 13, 5867, https://doi.org/10.1038/s41467-022-33583-4, 2022.

Drysdale, R. N., Hellstrom, J. C., Zanchetta, G., Fallick, A. E., Sánchez Goñi, M. F., Couchoud, I., McDonald, J., Maas, R.,
Lohmann, G., and Isola, I.: Evidence for Obliquity Forcing of Glacial Termination II, Science, 325, 1527–1531,
https://doi.org/10.1126/science.1170371, 2009.

Du, X., Russell, J. M., Liu, Z., Otto-Bliesner, B. L., Gao, Y., Zhu, C., Oppo, D. W., Mohtadi, M., Yan, Y., Galy, V. V., and
He, C.: Deglacial trends in Indo-Pacific warm pool hydroclimate in an isotope-enabled Earth system model and implications
for isotope-based paleoclimate reconstructions, Quat. Sci. Rev., 270, 107188,
https://doi.org/10.1016/j.quascirev.2021.107188, 2021.

Duplessy, J. C., Bard, E., Arnold, M., Shackleton, N. J., Duprat, J., and Labeyrie, L.: How fast did the ocean—atmosphere
system run during the last deglaciation?, Earth Planet. Sci. Lett., 103, 27–40, https://doi.org/10.1016/0012-821X(91)90147-
A, 1991.

Duplessy, J.-C., Labeyrie, L., and Waelbroeck, C.: Constraints on the ocean oxygen isotopic enrichment between the Last
Glacial Maximum and the Holocene: Paleoceanographic implications, Quat. Sci. Rev., 21, 315–330,
https://doi.org/10.1016/S0277-3791(01)00107-X, 2002.

Elderfield, H., Ferretti, P., Greaves, M., Crowhurst, S., McCave, I. N., Hodell, D., and Piotrowski, A. M.: Evolution of
Ocean Temperature and Ice Volume Through the Mid-Pleistocene Climate Transition, Science, 337, 704–709,
https://doi.org/10.1126/science.1221294, 2012.

Fairchild, I. J. and Treble, P. C.: Trace elements in speleothems as recorders of environmental change, Quat. Sci. Rev., 28,
449–468, https://doi.org/10.1016/j.quascirev.2008.11.007, 2009.

Fairchild, I. J., Borsato, A., Tooth, A. F., Frisia, S., Hawkesworth, C. J., Huang, Y., McDermott, F., and Spiro, B.: Controls
on trace element (Sr–Mg) compositions of carbonate cave waters: implications for speleothem climatic records, Chem.
Geol., 166, 255–269, https://doi.org/10.1016/S0009-2541(99)00216-8, 2000.

Fischer, H., Wahlen, M., Smith, J., Mastroianni, D., and Deck, B.: Ice Core Records of Atmospheric CO2 Around the Last
Three Glacial Terminations, Science, 283, 1712–1714, https://doi.org/10.1126/science.283.5408.1712, 1999.

Frumkin, A., Ford, D. C., and Schwarcz, H. P.: Continental Oxygen Isotopic Record of the Last 170,000 Years in Jerusalem,
Quat. Res., 51, 317–327, https://doi.org/10.1006/qres.1998.2031, 1999.

Gallup, C. D., Cheng, H., Taylor, F. W., and Edwards, R. L.: Direct Determination of the Timing of Sea Level Change
During Termination II, Science, 295, 310–313, https://doi.org/10.1126/science.1065494, 2002.

Gat, J. R.: OXYGEN AND HYDROGEN ISOTOPES IN THE HYDROLOGIC CYCLE, Annu. Rev. Earth Planet. Sci., 24,
225–262, https://doi.org/10.1146/annurev.earth.24.1.225, 1996.

Genty, D., Baker, A., Massault, M., Proctor, C., Gilmour, M., Pons-Branchu, E., and Hamelin, B.: Dead carbon in
stalagmites: carbonate bedrock paleodissolution vs. ageing of soil organic matter. Implications for 13C variations in
speleothems, Geochim. Cosmochim. Acta, 65, 3443–3457, https://doi.org/10.1016/S0016-7037(01)00697-4, 2001.



Genty, D., Blamart, D., Ouahdi, R., Gilmour, M., Baker, A., Jouzel, J., and Van-Exter, S.: Precise dating of Dansgaard–Oeschger climate oscillations in western Europe from stalagmite data, Nature, 421, 833–837, https://doi.org/10.1038/nature01391, 2003.

Genty, D., Blamart, D., Ghaleb, B., Plagnes, V., Causse, Ch., Bakalowicz, M., Zouari, K., Chkir, N., Hellstrom, J., Wainer, K., and Bourges, F.: Timing and dynamics of the last deglaciation from European and North African δ13C stalagmite
profiles—comparison with Chinese and South Hemisphere stalagmites, Quat. Sci. Rev., 25, 2118–2142, https://doi.org/10.1016/j.quascirev.2006.01.030, 2006.

Grant, K. M., Rohling, E. J., Bar-Matthews, M., Ayalon, A., Medina-Elizalde, M., Ramsey, C. B., Satow, C., and Roberts, A. P.: Rapid coupling between ice volume and polar temperature over the past 150,000 years, Nature, 491, 744–747, https://doi.org/10.1038/nature11593, 2012.

Grant, K. M., Rohling, E. J., Ramsey, C. B., Cheng, H., Edwards, R. L., Florindo, F., Heslop, D., Marra, F., Roberts, A. P., Tamisiea, M. E., and Williams, F.: Sea-level variability over five glacial cycles, Nat. Commun., 5, 5076, https://doi.org/10.1038/ncomms6076, 2014.

Hardt, B., Rowe, H. D., Springer, G. S., Cheng, H., and Edwards, R. L.: The seasonality of east central North American precipitation based on three coeval Holocene speleothems from southern West Virginia, Earth Planet. Sci. Lett., 295, 342–
348, https://doi.org/10.1016/j.epsl.2010.04.002, 2010.

Harmon, R. S., Schwarcz, H. P., and O'Neil, J. R.: D/H ratios in speleothem fluid inclusions: A guide to variations in the isotopic composition of meteoric precipitation?, Earth Planet. Sci. Lett., 42, 254–266, https://doi.org/10.1016/0012-821X(79)90033-5, 1979.

Hatvani, I. G., Kern, Z., Tanos, P., Wilhelm, M., Lechleitner, F. A., and Kaushal, N.: The SISAL webApp: exploring the
speleothem climate and environmental archives of the world, Quat. Res., 118, 211–217, https://doi.org/10.1017/qua.2023.39, 2024.

He, C., Liu, Z., Otto-Bliesner, B. L., Brady, E. C., Zhu, C., Tomas, R., Clark, P. U., Zhu, J., Jahn, A., Gu, S., Zhang, J., Nusbaumer, J., Noone, D., Cheng, H., Wang, Y., Yan, M., and Bao, Y.: Hydroclimate footprint of pan-Asian monsoon water isotope during the last deglaciation, Sci. Adv., 7, eabe2611, https://doi.org/10.1126/sciadv.abe2611, 2021.

Held, F., Cheng, H., Edwards, R. L., Tüysüz, O., Koç, K., and Fleitmann, D.: Dansgaard-Oeschger cycles of the penultimate and last glacial period recorded in stalagmites from Türkiye, Nat. Commun., 15, 1183, https://doi.org/10.1038/s41467-024-45507-5, 2024.

Hobart, B., Lisiecki, L. E., Rand, D., Lee, T., and Lawrence, C. E.: Late Pleistocene 100-kyr glacial cycles paced by precession forcing of summer insolation, Nat. Geosci., 16, 717–722, https://doi.org/10.1038/s41561-023-01235-x, 2023.

Ivanovic, R. F., Gregoire, L. J., Kageyama, M., Roche, D. M., Valdes, P. J., Burke, A., Drummond, R., Peltier, W. R., and Tarasov, L.: Transient climate simulations of the deglaciation 21–9 thousand years before present (version 1) – PMIP4 Core experiment design and boundary conditions, Geosci. Model Dev., 9, 2563–2587, https://doi.org/10.5194/gmd-9-2563-2016, 2016.

Jackson, L. C. and Wood, R. A.: Timescales of AMOC decline in response to fresh water forcing, Clim. Dyn., 51, 1333–
1350, https://doi.org/10.1007/s00382-017-3957-6, 2018.



Johnson, K. R., Hu, C., Belshaw, N. S., and Henderson, G. M.: Seasonal trace-element and stable-isotope variations in a Chinese speleothem: The potential for high-resolution paleomonsoon reconstruction, Earth Planet. Sci. Lett., 244, 394–407, https://doi.org/10.1016/j.epsl.2006.01.064, 2006.

Kathayat, G., Cheng, H., Sinha, A., Spötl, C., Edwards, R. L., Zhang, H., Li, X., Yi, L., Ning, Y., Cai, Y., Lui, W. L., and Breitenbach, S. F. M.: Indian monsoon variability on millennial-orbital timescales, Sci. Rep., 6, 24374, https://doi.org/10.1038/srep24374, 2016.

Kaushal, N., Lechleitner, F. A., Wilhelm, M., Azennoud, K., Bühler, J. C., Braun, K., Ait Brahim, Y., Baker, A., Burstyn, Y., Comas-Bru, L., Fohlmeister, J., Goldsmith, Y., Harrison, S. P., Hatvani, I. G., Rehfeld, K., Ritzau, M., Skiba, V., Stoll, H. M., Szűcs, J. G., Tanos, P., Treble, P. C., Azevedo, V., Baker, J. L., Borsato, A., Chawchai, S., Columbu, A., Endres, L., Hu, J., Kern, Z., Kimbrough, A., Koç, K., Markowska, M., Martrat, B., Masood Ahmad, S., Nehme, C., Novello, V. F., Pérez-Mejías, C., Ruan, J., Sekhon, N., Sinha, N., Tadros, C. V., Tiger, B. H., Warken, S., Wolf, A., Zhang, H., and SISAL Working Group members: SISALv3: a global speleothem stable isotope and trace element database, Earth Syst. Sci. Data, 16, 1933–1963, https://doi.org/10.5194/essd-16-1933-2024, 2024a.

Kaushal, N., Lechleitner, F. A., Wilhelm, M., and Members, S. W. G.: SISALv3: Speleothem Isotopes Synthesis and AnaLysis database version 3.0, 2024b.

Kawamura, K., Parrenin, F., Lisiecki, L., Uemura, R., Vimeux, F., Severinghaus, J. P., Hutterli, M. A., Nakazawa, T., Aoki, S., Jouzel, J., Raymo, M. E., Matsumoto, K., Nakata, H., Motoyama, H., Fujita, S., Goto-Azuma, K., Fujii, Y., and Watanabe, O.: Northern Hemisphere forcing of climatic cycles in Antarctica over the past 360,000 years, Nature, 448, 912–916, https://doi.org/10.1038/nature06015, 2007.

Kelly, M. J., Edwards, R. L., Cheng, H., Yuan, D., Cai, Y., Zhang, M., Lin, Y., and An, Z.: High resolution characterization of the Asian Monsoon between 146,000 and 99,000 years B.P. from Dongge Cave, China and global correlation of events surrounding Termination II, Palaeogeogr. Palaeoclimatol. Palaeoecol., 236, 20–38, https://doi.org/10.1016/j.palaeo.2005.11.042, 2006.

Kim, S.-T., Mucci, A., and Taylor, B. E.: Phosphoric acid fractionation factors for calcite and aragonite between 25 and 75 °C: Revisited, Chem. Geol., 246, 135–146, https://doi.org/10.1016/j.chemgeo.2007.08.005, 2007.

Kimbrough, A. K., Gagan, M. K., Dunbar, G. B., Hantoro, W. S., Shen, C.-C., Hu, H.-M., Cheng, H., Edwards, R. L., Rifai, H., and Suwargadi, B. W.: Multi-proxy validation of glacial-interglacial rainfall variations in southwest Sulawesi, Commun. Earth Environ., 4, 1–13, https://doi.org/10.1038/s43247-023-00873-8, 2023.

Koltai, G., Spötl, C., Shen, C.-C., Wu, C.-C., Rao, Z., Palcsu, L., Kele, S., Surányi, G., and Bárány-Kevei, I.: A penultimate glacial climate record from southern Hungary, J. Quat. Sci., 32, 946–956, https://doi.org/10.1002/jqs.2968, 2017.

Lachniet, M. S., Denniston, R. F., Asmerom, Y., and Polyak, V. J.: Orbital control of western North America atmospheric circulation and climate over two glacial cycles, Nat. Commun., 5, 3805, https://doi.org/10.1038/ncomms4805, 2014.

Lamy, F., Kaiser, J., Arz, H. W., Hebbeln, D., Ninnemann, U., Timm, O., Timmermann, A., and Toggweiler, J. R.: Modulation of the bipolar seesaw in the Southeast Pacific during Termination 1, Earth Planet. Sci. Lett., 259, 400–413, https://doi.org/10.1016/j.epsl.2007.04.040, 2007.

Lea, D. W., Pak, D. K., Peterson, L. C., and Hughen, K. A.: Synchroneity of Tropical and High-Latitude Atlantic Temperatures over the Last Glacial Termination, Science, 301, 1361–1364, https://doi.org/10.1126/science.1088470, 2003.





Lechleitner, F. A., Day, C. C., Kost, O., Wilhelm, M., Haghipour, N., Henderson, G. M., and Stoll, H. M.: Stalagmite carbon isotopes suggest deglacial increase in soil respiration in western Europe driven by temperature change, Clim. Past, 17, 1903–1918, https://doi.org/10.5194/cp-17-1903-2021, 2021.

Lisiecki, L. E. and Raymo, M. E.: Plio–Pleistocene climate evolution: trends and transitions in glacial cycle dynamics, Quat. Sci. Rev., 26, 56–69, https://doi.org/10.1016/j.quascirev.2006.09.005, 2007.

Liu, X., Rao, Z., Zhang, X., Huang, W., Chen, J., and Chen, F.: Variations in the oxygen isotopic composition of precipitation in the Tianshan Mountains region and their significance for the Westerly circulation, J. Geogr. Sci., 25, 801–816, https://doi.org/10.1007/s11442-015-1203-x, 2015.

Liu, Z., Otto-Bliesner, B. L., He, F., Brady, E. C., Tomas, R., Clark, P. U., Carlson, A. E., Lynch-Stieglitz, J., Curry, W., Brook, E., Erickson, D., Jacob, R., Kutzbach, J., and Cheng, J.: Transient Simulation of Last Deglaciation with a New Mechanism for Bølling-Allerød Warming, Science, 325, 310–314, https://doi.org/10.1126/science.1171041, 2009.

Liu, Z., Wen, X., Brady, E. C., Otto-Bliesner, B., Yu, G., Lu, H., Cheng, H., Wang, Y., Zheng, W., Ding, Y., Edwards, R. L., Cheng, J., Liu, W., and Yang, H.: Chinese cave records and the East Asia Summer Monsoon, Quat. Sci. Rev., 83, 115–128, https://doi.org/10.1016/j.quascirev.2013.10.021, 2014.

Maher, B. A.: Holocene variability of the East Asian summer monsoon from Chinese cave records: a re-assessment, The Holocene, 18, 861–866, https://doi.org/10.1177/0959683608095569, 2008.

Matthews, A., Affek, H. P., Ayalon, A., Vonhof, H. B., and Bar-Matthews, M.: Eastern Mediterranean climate change deduced from the Soreq Cave fluid inclusion stable isotopes and carbonate clumped isotopes record of the last 160 ka, Quat. Sci. Rev., 272, 107223, https://doi.org/10.1016/j.quascirev.2021.107223, 2021.

Meckler, A. N., Clarkson, M. O., Cobb, K. M., Sodemann, H., and Adkins, J. F.: Interglacial Hydroclimate in the Tropical West Pacific Through the Late Pleistocene, Science, 336, 1301–1304, https://doi.org/10.1126/science.1218340, 2012.

Meckler, A. N., Affolter, S., Dublyansky, Y. V., Krüger, Y., Vogel, N., Bernasconi, S. M., Frenz, M., Kipfer, R., Leuenberger, M., Spötl, C., Carolin, S., Cobb, K. M., Moerman, J., Adkins, J. F., and Fleitmann, D.: Glacial–interglacial temperature change in the tropical West Pacific: A comparison of stalagmite-based paleo-thermometers, Quat. Sci. Rev., 127, 90–116, https://doi.org/10.1016/j.quascirev.2015.06.015, 2015.

Menviel, L., Capron, E., Govin, A., Dutton, A., Tarasov, L., Abe-Ouchi, A., Drysdale, R. N., Gibbard, P. L., Gregoire, L., He, F., Ivanovic, R. F., Kageyama, M., Kawamura, K., Landais, A., Otto-Bliesner, B. L., Oyabu, I., Tzedakis, P. C., Wolff, E., and Zhang, X.: The penultimate deglaciation: protocol for Paleoclimate Modelling Intercomparison Project (PMIP) phase 4 transient numerical simulations between 140 and 127 ka, version 1.0, Geosci. Model Dev., 12, 3649–3685, https://doi.org/10.5194/gmd-12-3649-2019, 2019.

Mitsui, T., Tzedakis, P. C., and Wolff, E. W.: Insolation evolution and ice volume legacies determine interglacial and glacial intensity, Clim. Past, 18, 1983–1996, https://doi.org/10.5194/cp-18-1983-2022, 2022.

Moseley, G. E., Spötl, C., Cheng, H., Boch, R., Min, A., and Edwards, R. L.: Termination-II interstadial/stadial climate change recorded in two stalagmites from the north European Alps, Quat. Sci. Rev., 127, 229–239, https://doi.org/10.1016/j.quascirev.2015.07.012, 2015.

Nehme, C., Kluge, T., Verheyden, S., Nader, F., Charalambidou, I., Weissbach, T., Gucel, S., Cheng, H., Edwards, R. L., Satterfield, L., Eiche, E., and Claeys, P.: Speleothem record from Pentadactylos cave (Cyprus): new insights into climatic



variations during MIS 6 and MIS 5 in the Eastern Mediterranean, Quat. Sci. Rev., 250, 106663, https://doi.org/10.1016/j.quascirev.2020.106663, 2020.

Obase, T., Abe-Ouchi, A., and Saito, F.: Abrupt climate changes in the last two deglaciations simulated with different Northern ice sheet discharge and insolation, Sci. Rep., 11, 22359, https://doi.org/10.1038/s41598-021-01651-2, 2021.

Parker, S. E., Harrison, S. P., Comas-Bru, L., Kaushal, N., LeGrande, A. N., and Werner, M.: A data–model approach to
interpreting speleothem oxygen isotope records from monsoon regions, Clim. Past, 17, 1119–1138, https://doi.org/10.5194/cp-17-1119-2021, 2021.

Pausata, F. S. R., Battisti, D. S., Nisancioglu, K. H., and Bitz, C. M.: Chinese stalagmite δ18O controlled by changes in the Indian monsoon during a simulated Heinrich event, Nat. Geosci., 4, 474–480, https://doi.org/10.1038/ngeo1169, 2011.

Pérez-Mejías, C., Moreno, A., Sancho, C., Bartolomé, M., Stoll, H., Cacho, I., Cheng, H., and Edwards, R. L.: Abrupt
climate changes during Termination III in Southern Europe, Proc. Natl. Acad. Sci., 114, 10047–10052, https://doi.org/10.1073/pnas.1619615114, 2017.

Raymo, M. E., Lisiecki, L. E., and Nisancioglu, K. H.: Plio-Pleistocene Ice Volume, Antarctic Climate, and the Global δ18O Record, Science, 313, 492–495, https://doi.org/10.1126/science.1123296, 2006.

Rehfeld, K. and Bühler, J.: Age-depth model ensembles for SISAL v3 speleothem records,
https://doi.org/10.5281/zenodo.10726619, 2024.

Rehfeld, K., Roesch, C., Comas-Bru, L., and Amirnezhad-Mozhdehi, S.: Age-depth model ensembles for SISAL v2 speleothem records (1.0), https://doi.org/10.5281/zenodo.3816804, 2020.

Risi, C., Bony, S., and Vimeux, F.: Influence of convective processes on the isotopic composition (δ18O and δD) of precipitation and water vapor in the tropics: 2. Physical interpretation of the amount effect, J. Geophys. Res. Atmospheres,
113, https://doi.org/10.1029/2008JD009943, 2008.

Rozanski, K., Araguás-Araguás, L., and Gonfiantini, R.: Isotopic Patterns in Modern Global Precipitation, in: Geophysical Monograph Series, edited by: Swart, P. K., Lohmann, K. C., Mckenzie, J., and Savin, S., American Geophysical Union, Washington, D. C., 1–36, https://doi.org/10.1029/GM078p0001, 2013.

Schrag, D. P., Hampt, G., and Murray, D. W.: Pore Fluid Constraints on the Temperature and Oxygen Isotopic Composition
of the Glacial Ocean, Science, 272, 1930–1932, https://doi.org/10.1126/science.272.5270.1930, 1996.

Schrag, D. P., Adkins, J. F., McIntyre, K., Alexander, J. L., Hodell, D. A., Charles, C. D., and McManus, J. F.: The oxygen isotopic composition of seawater during the Last Glacial Maximum, Quat. Sci. Rev., 21, 331–342, https://doi.org/10.1016/S0277-3791(01)00110-X, 2002.

Shakun, J. D., Burns, S. J., Clark, P. U., Cheng, H., and Edwards, R. L.: Milankovitch-paced Termination II in a Nevada
speleothem?, Geophys. Res. Lett., 38, https://doi.org/10.1029/2011GL048560, 2011.

Shakun, J. D., Lea, D. W., Lisiecki, L. E., and Raymo, M. E.: An 800-kyr record of global surface ocean $\delta^{18}O$ and implications for ice volume-temperature coupling, Earth Planet. Sci. Lett., 426, 58–68, https://doi.org/10.1016/j.epsl.2015.05.042, 2015.



Sinha, A., Kathayat, G., Cheng, H., Breitenbach, S. F. M., Berkelhammer, M., Mudelsee, M., Biswas, J., and Edwards, R. L.: Trends and oscillations in the Indian summer monsoon rainfall over the last two millennia, Nat. Commun., 6, 6309, https://doi.org/10.1038/ncomms7309, 2015.

Sinha, N., Timmermann, A., Lee, S.-S., Jo, K.-N., Wassenburg, J., Cleary, D., and Yun, K.-S.: Orbital-scale Dynamics of the Eastern Asian Summer Monsoon, https://doi.org/10.21203/rs.3.rs-3691295/v1, 3 April 2024.

Sinnl, G., Adolphi, F., Christl, M., Welten, K. C., Woodruff, T., Caffee, M., Svensson, A., Muscheler, R., and Rasmussen, S. O.: Synchronizing ice-core and U / Th timescales in the Last Glacial Maximum using Hulu Cave $^{14}$C and new $^{10}$Be measurements from Greenland and Antarctica, Clim. Past, 19, 1153–1175, https://doi.org/10.5194/cp-19-1153-2023, 2023.

Spratt, R. M. and Lisiecki, L. E.: A Late Pleistocene sea level stack, Clim. Past, 12, 1079–1092, https://doi.org/10.5194/cp-1080    12-1079-2016, 2016.

Stern, J. V. and Lisiecki, L. E.: Termination 1 timing in radiocarbon-dated regional benthic δ18O stacks, Paleoceanography, 29, 1127–1142, https://doi.org/10.1002/2014PA002700, 2014.

Stoll, H. M., Cacho, I., Gasson, E., Sliwinski, J., Kost, O., Moreno, A., Iglesias, M., Torner, J., Perez-Mejias, C., Haghipour, N., Cheng, H., and Edwards, R. L.: Rapid northern hemisphere ice sheet melting during the penultimate deglaciation, Nat. 1085    Commun., 13, 3819, https://doi.org/10.1038/s41467-022-31619-3, 2022.

Stoll, H. M., Day, C., Lechleitner, F., Kost, O., Endres, L., Sliwinski, J., Pérez-Mejías, C., Cheng, H., and Scholz, D.: Distinguishing the combined vegetation and soil component of $\delta^{13}$C variation in speleothem records from subsequent degassing and prior calcite precipitation effects, Clim. Past, 19, 2423–2444, https://doi.org/10.5194/cp-19-2423-2023, 2023.

Suwa, M. and Bender, M. L.: Chronology of the Vostok ice core constrained by O2/N2 ratios of occluded air, and its 1090    implication for the Vostok climate records, Quat. Sci. Rev., 27, 1093–1106, https://doi.org/10.1016/j.quascirev.2008.02.017, 2008.

Tabor, C., Lofverstrom, M., Oster, J., Wortham, B., de Wet, C., Montañez, I., Rhoades, A., Zarzycki, C., He, C., and Liu, Z.: A mechanistic understanding of oxygen isotopic changes in the Western United States at the Last Glacial Maximum, Quat. Sci. Rev., 274, 107255, https://doi.org/10.1016/j.quascirev.2021.107255, 2021.

Thomas, E. K., Clemens, S. C., Prell, W. L., Herbert, T. D., Huang, Y., Liu, Z., Sinninghe Damsté, J. S., Sun, Y., and Wen, X.: Temperature and leaf wax δ2H records demonstrate seasonal and regional controls on Asian monsoon proxies, Geology, 42, 1075–1078, https://doi.org/10.1130/G36289.1, 2014.

Tierney, J. E., Lewis, S. C., Cook, B. I., LeGrande, A. N., and Schmidt, G. A.: Model, proxy and isotopic perspectives on the East African Humid Period, Earth Planet. Sci. Lett., 307, 103–112, https://doi.org/10.1016/j.epsl.2011.04.038, 2011.

Tzedakis, P. C., Crucifix, M., Mitsui, T., and Wolff, E. W.: A simple rule to determine which insolation cycles lead to interglacials, Nature, 542, 427–432, https://doi.org/10.1038/nature21364, 2017.

Tzedakis, P. C., Drysdale, R. N., Margari, V., Skinner, L. C., Menviel, L., Rhodes, R. H., Taschetto, A. S., Hodell, D. A., Crowhurst, S. J., Hellstrom, J. C., Fallick, A. E., Grimalt, J. O., McManus, J. F., Martrat, B., Mokeddem, Z., Parrenin, F., Regattieri, E., Roe, K., and Zanchetta, G.: Enhanced climate instability in the North Atlantic and southern Europe during the 1105    Last Interglacial, Nat. Commun., 9, 4235, https://doi.org/10.1038/s41467-018-06683-3, 2018.



Waelbroeck, C., Labeyrie, L., Michel, E., Duplessy, J. C., McManus, J. F., Lambeck, K., Balbon, E., and Labracherie, M.: Sea-level and deep water temperature changes derived from benthic foraminifera isotopic records, Quat. Sci. Rev., 21, 295–305, https://doi.org/10.1016/S0277-3791(01)00101-9, 2002.

Waelbroeck, C., Kiefer, T., Dokken, T., Chen, M.-T., Spero, H. J., Jung, S., Weinelt, M., Kucera, M., and Paul, A.: Constraints on surface seawater oxygen isotope change between the Last Glacial Maximum and the Late Holocene, Quat. Sci. Rev., 105, 102–111, https://doi.org/10.1016/j.quascirev.2014.09.020, 2014.

Wainer, K., Genty, D., Blamart, D., Daëron, M., Bar-Matthews, M., Vonhof, H., Dublyansky, Y., Pons-Branchu, E., Thomas, L., van Calsteren, P., Quinif, Y., and Caillon, N.: Speleothem record of the last 180 ka in Villars cave (SW France): Investigation of a large δ18O shift between MIS6 and MIS5, Quat. Sci. Rev., 30, 130–146, https://doi.org/10.1016/j.quascirev.2010.07.004, 2011.

Wassenburg, J. A., Vonhof, H. B., Cheng, H., Martínez-García, A., Ebner, P.-R., Li, X., Zhang, H., Sha, L., Tian, Y., Edwards, R. L., Fiebig, J., and Haug, G. H.: Penultimate deglaciation Asian monsoon response to North Atlantic circulation collapse, Nat. Geosci., 14, 937–941, https://doi.org/10.1038/s41561-021-00851-9, 2021.

Wegwerth, A., Dellwig, O., Wulf, S., Plessen, B., Kleinhanns, I. C., Nowaczyk, N. R., Jiabo, L., and Arz, H. W.: Major hydrological shifts in the Black Sea "Lake" in response to ice sheet collapses during MIS 6 (130–184 ka BP), Quat. Sci. Rev., 219, 126–144, https://doi.org/10.1016/j.quascirev.2019.07.008, 2019.

Wendt, K. A., Li, X., Edwards, R. L., Cheng, H., and Spötl, C.: Precise timing of MIS 7 substages from the Austrian Alps, Clim. Past, 17, 1443–1454, https://doi.org/10.5194/cp-17-1443-2021, 2021.

Wilcox, P. S., Honiat, C., Trüssel, M., Edwards, R. L., and Spötl, C.: Exceptional warmth and climate instability occurred in the European Alps during the Last Interglacial period, Commun. Earth Environ., 1, 1–6, https://doi.org/10.1038/s43247-020-00063-w, 2020.

Zhang, H., Ait Brahim, Y., Li, H., Zhao, J., Kathayat, G., Tian, Y., Baker, J., Wang, J., Zhang, F., Ning, Y., Edwards, R. L., and Cheng, H.: The Asian Summer Monsoon: Teleconnections and Forcing Mechanisms—A Review from Chinese Speleothem δ18O Records, Quaternary, 2, 26, https://doi.org/10.3390/quat2030026, 2019.

Zhao, Y. and Harrison, S. P.: Mid-Holocene monsoons: a multi-model analysis of the inter-hemispheric differences in the responses to orbital forcing and ocean feedbacks, Clim. Dyn., 39, 1457–1487, https://doi.org/10.1007/s00382-011-1193-z, 2012.