# Peer review of "Perspective on ice age Terminations from absolute chronologies provided by global speleothem records"

_Climate of the Past, 2024_

## Author Comment (AC1)

RC1: 'Comment on cp-2024-37', Jasper Wassenburg, 23 Jun 2024

Review Kaushal et al. "Perspective on ice age Terminations from absolute chronologies provided by global speleothem records".

General comments:

The presented manuscript compiles a global dataset of speleothem ice age Termination records for TII to TV with the purpose of describing chronological sequences of events, discuss differences and similarities between Terminations and the effects of different ice volume corrections. I believe this to be a very valuable contribution that clearly outlines future directions and targets for work on ice age Terminations. In particular, the tuning of other climate archives to speleothem proxies highlights the many purposes of speleothem records. In this regard the authors could, however, indicate some of the potential pitfalls more clearly. For example, correlating climate archives over large distances should be done with caution and only if proven that the different climate parameters respond to the same forcings without delay. The authors suggestion to use isotope enabled climate models for this purpose is indeed a critical one. Overall, the conclusions are well supported by the discussion. I'm looking forward to see this work published with a few revisions.

Dear Prof. Wassenburg,

Thank you for reviewing the article. And thank you for your kind and articulate comments valuing the work presented in this publication. We greatly appreciate your suggestions. We have addressed your comments below. These suggestions will certainly improve the quality of this manuscript.

Specific comments:

My main comment only concerns the structure of the manuscript concerning seawater isotope corrections / ice volume corrections. Right now there is one subchapter (2.2.) devoted to "ice volume corrections". Within this chapter also P-E changes and its effect on surface seawater d18O is discussed. I believe this subchapter should be named "sea surface d18O corrections" instead, because ice volume directly effects sea surface d18O as well.

Throughout the manuscript there is an ongoing discussion about sea surface d18O corrections. I think it would streamline the paper if everything concerning these corrections could be discussed in chapter 2.2, which ends with a clear conclusion on how every speleothem d18O record is corrected. This also means to move chapter 3.1. to chapter 2.

Thank you for this suggestion. This is something we have considered before as well. This is a long paper with a lot of text, and it took us multiple presentations at workshops (INQUA, EGU) to figure out the clearest way of presenting all the information. We settled on the current format to make clear which datasets had been extracted or modified by us in this manuscript in section 2 versus presenting interpretations in section 3.

Section (2) of the manuscript deals with 'data processing'. So that subsection 2.1 addresses which data was extracted and used, subsection 2.2 addresses which datasets have been modified to accommodate for ice-volume corrections and subsection 2.3 addresses which datasets have been modified to accommodate for degassing corrections.

Section (3) examines the different climate aspects that have been recorded by speleothems. So that subsection 3.1 addresses records of surface ocean freshening and so on

Subsection 2.2 examines which ice volume corrections are available and how these corrections have been made. And subsection 3.1 provides interpretations of the NISA record (which has been used to make corrections as detailed in subsection 2.2) as well as the Villars and Sofular cave records. We will add the Corchia cave record to subsection 3.1 based on Referee 2's comments as well.

However, we realise that subsection 2.2 does not detail which records have been corrected which would make this subsection clearer. We will add a sentence to Line 240 as follows (additional sentence highlighted in bold):

Wherever correction for changing $\delta$18Oseawater is implemented, the speleothem data have been binned to 1000, 250 and 125 years respectively to accommodate uncertainties in the speleothem chronology (Supp. Fig. 4). The change in $\delta$18Oseawater as a result of freshening has simply been subtracted from the speleothem $\delta$18O in these bins. Since the uncertainty on sea level curves is much greater than the uncertainty on speleothem age-depth models, only the uncertainty on sea level curves has been considered in these plots. **The ice volume 'corrected' Termination II Abaliget (ABA_1), Sieben Hengste (7H-12), Schneckenloch (SCH-5)and Corchia (CC-5_2018) cave records have been used for further interpretation in the main manuscript and the corrected records are indicated by the Y-axis labels 'd18Ocorr' in Figure 4. The absence of an equivalent absolute dated record of North Atlantic $\delta$18Oseawater evolution in prior Terminations precludes regional correction of temperature equivalent European records in TIII or older at this time, instead the global ice volume correction has been applied to these records (Supp. Fig. 4).**

Lines 270 – 271: Considering the importance of the NISA d18O record for surface seawater isotope corrections I think it would be helpful to provide more background information why the NISA d18O can be used for this purpose as opposed to only referring to Stoll et al. (2022). Could you please comment on the potential temperature effect on the water to calcite isotope fractionation? A simple sentence that includes the effect of rainfall isotope d18O vs temperature and the cave air temperature water to calcite isotope fractionation would be sufficient. Then the reader who wonders why cave air temperature does not affect CaCO3d18O will readily understand this interpretation as well.

 This is a really good suggestion! We will add the following text to Lines 270:

Changes in the $\delta$18Oseawater of the moisture source for caves and drip waters may be the dominant signal in speleothem $\delta$18O in some settings. The $\delta$18O in speleothems from coastal caves in Northwest Spain (NISA) is dominantly controlled by the $\delta$18Oseawater of the eastern North Atlantic, as documented in comparison with independently dated $\delta$18Oseawater records from foraminifera over TI (Stoll et al., 2022). **Rainfall monitoring at this cave location shows that the slight decrease in rainfall d18O with decreasing temperature appears to be of similar magnitude but opposite in sign to the temperature-dependant fractionation between drip water and calcite leaving the $\delta$18Oseawater of the North Atlantic Ocean as the main signal expressed by the speleothems (Stoll et al., 2015; Stoll et al., 2022).** Because of its proximity to the source of meltwater release, the $\delta$18Oseawater of the surface ocean in the North Atlantic experiences a higher amplitude change in $\delta$18Oseawater across a glacial cycle, and may record transient millennial scale events in the $\delta$18Oseawater. Over TII, NISA speleothems provide a record of the timing of deglacial freshening of

the eastern North Atlantic with a $\delta$18O amplitude of ~2.5 ‰ (Supp. Fig. 6). Other coastal caves on the Atlantic margin, such as Villars Cave (Supp. Fig. 2), may also be dominated by the change in isotopic composition of the North Atlantic.

Line 325 (and 343 – 345): Temperature is reconstructed with different proxies. Some may record cave air temperature, some record a vegetation - temperature driven d13C, and others record atmospheric air temperature with d2H or CaCO3 d18O. The seasons that are recorded by the different proxies may have a large impact on the reconstructed temperature amplitude, it would be good to mention and discuss this in more detail.

This comment has been addressed in the Line 325 technical comment below.

Chapter 5.3. Nice overview of how speleothem chronologies could potentially be used as tuning targets for climate archives that lack absolute chronologies. I do believe that this chapter could benefit if the potential pitfalls would be described. Please caution against tuning between records over long distances that are not necessarily part of the same systems.

 We propose to end that paragraph (line 690) with a clarifying statement:

Yet even when such distant correlation is based on strong common drivers of distal signals, regional climate processes which are independent of the common processes may add additional variability to each record which complicates robust tuning. As more absolute dated speleothem records emerge, it will be possible to more rigorously evaluate the fidelity of these long-distance teleconnections on varying timescales.

Technical comments:

Line 12: should be "largest amplitude global climate"

Thank you. We will add the word 'global' to this sentence.

Line 15: "a sequence of feedbacks" does not seem correct as a feedback is a consequence of an event that reinforces (positive) or buffers (negative) the effects of the event itself. Maybe rewrite to: "the sequence of millennial events, their climate feedbacks and rates of change".

That's a good point. We will make this change.

Line 19: "and unlike proxies in other archives like ice or marine cores," I would delete this part. Ice cores over the world cannot be interpreted similar, for example if you compare a d18O ice from the Andes mountain range it may not be related to temperature as it is in the NGRIP ice core. Also marine sediments have proxies that may be interpreted differently around the world: In the Mediterranean d18O of surface dwelling foraminifera may be a P-E signal, whereas it might be dominated by temperature in regions where P-E is less dominant (polar regions?). I would rewrite it like this: "are encoded in a number of proxies, however, the climatic"

You are absolutely correct in the details. We would still like to flag the differences between marine and ice core d18O on the one hand and speleothem d18O on the other hand, particularly to

researchers who are not from the speleothem field, and we will clarify that we refer to **polar** ice core d18O-based temperature records and the **benthic foraminiferal** d18O-based sea level curves. In such records the d18O proxy, no matter the location, has the same overall interpretation, though of course subject to regional nuances. Whereas this is really not the case when it comes to speleothem d18O records. Speleothem d18O records from one location may track temperature-dependency of meteoric precipitation, and in another they may track source water changes. While we appreciate, and agree with you regarding the details, we would like to retain the wording that we currently have so that we can flag the larger differences.

Line 28: "maybe" should be "may be"

Thank you. We will make this correction.

Line 29: "IIA" should be "IIIA"

Thank you. We will make this correction.

Line 42: see comment on line 12

Thank you. We will make this change.

Line 51: see comment on line 15

Thank you. We will make this change.

Line 68: add reference "Lisiecki and Stern (2016)" they also use the EA speleothem record to tune the older part of the record.

Thank you. We will add this reference.

Line 143: "the timing temperature change," should read "the timing of temperature changes"?

Yes! Thank you. We will make this correction

Line 143: temperature reconstruction can be provided by multiple proxies, such as fluid inclusions d2H (Affolter et al., 2019), calcite-water d18O with fluid inclusion and calcite d18O, TEX86 (Levy et al., 2023; Wassenburg et al., 2021) as well as (dual) clumped isotopes (Bajnai et al., 2020; Wassenburg et al., 2021).

Thank you. We will add these details and references to Section 2 as well.

Figure 1: To give the reader an idea of the total nr of Termination records, it would be good to include all available Termination speleothem records in Figure 1. This also gives the reader an idea of how many records have been excluded by using the author's criteria and assess potential (if any) biases towards certain records or regions. The prioritized records could be indicated with different symbols or color as the ones that were left out.

This is a good suggestion. We will make this change. The modified maps have been given below.

[Figure]

[Figure]

[Figure]

**Termination III**

[Figure]

**Termination IV**

[Figure]

**Termination V**

Line 174: better reference is "Fohlmeister et al. (2018)" which is speleothem specific instead of "Kim et al., 2007" even though the difference between calcite and aragonite is similar, i.e. 0.8 permille.

This is a good suggestion. We will add the reference to Fohlmeister et al, 2018 as well.

Fohlmeister, Jens, et al. "Carbon and oxygen isotope fractionation in the water-calcite-aragonite system." *Geochimica et Cosmochimica Acta* 235 (2018): 127-139.

Line 293: Provided that kinetic offsets from isotope equilibrium do not change.

We acknowledge that variation in oxygen isotopic fractionation (and the potential influence of PCP on oxygen isotopes) is under discussion. We feel that line 293 is not the optimal place to add this detail, since indeed this affects interpretation of all oxygen isotope records in speleothems not only those in regions sensitive to temperature effects. Therefore, we propose to raise this as a point in the comparison of calcite d18O with fluid inclusion d18O in line 334:

Despite their lower temporal resolution, one advantage of fluid inclusion d18O measurements is that unlike d18O calcite records, the interpretation of fluid inclusion d18O does not require assumption of constant d18O water-calcite fractionation. The reliability of fluid inclusion analytical methods is improving with techniques to correct for analytical evaporation effects (Fernandez et al 2023).

Fernandez, A., Løland, M. H., Maccali, J., Krüger, Y., Vonhof, H. B., Sodemann, H., & Meckler, A. N. (2023). Characterization and correction of evaporative artifacts in speleothem fluid inclusion isotope analyses as applied to a stalagmite from Borneo. Geochemistry, Geophysics, Geosystems, 24, e2023GC010857. https://doi. org/10.1029/2023GC010857

Lines 303 - 304: See comment on lines 270 – 271.

We are now addressing this in lines 270-271.

Lines 307 - 309: Move to section 3.2. ice-volume corrections.

This is a good suggestion. Thank you. As detailed in the first comment, we would prefer to retain all corrections made in Section 2. The last paragraph in Section 2.2. in the revised manuscript will now read as follows:

Wherever correction for changing $\delta$18Oseawater is implemented, the speleothem data have been binned to 1000, 250 and 125 years respectively to accommodate uncertainties in the speleothem chronology (Supp. Fig. 4). The change in $\delta$18Oseawater as a result of freshening has simply been subtracted from the speleothem $\delta$18O in these bins. Since the uncertainty on sea level curves is much greater than the uncertainty on speleothem age-depth models, only the uncertainty on sea level curves has been considered in these plots. The ice volume 'corrected' Termination II Abaliget (ABA_1), Sieben Hengste (7H-12), Schneckenloch (SCH-5)and Corchia (CC-5_2018) cave records have been used for further interpretation in the main manuscript and the corrected records are indicated by the Y-axis labels 'd18Ocorr' in Figure 4. The absence of an equivalent absolute dated record of North Atlantic $\delta$18Oseawater evolution in prior Terminations precludes regional

correction of temperature equivalent European records in TIII or older at this time, instead the global ice volume correction has been applied to these records (Supp. Fig. 4).

Line 325 (and 343 – 345): Temperature is reconstructed with different proxies. Some may record cave air temperature, some record a vegetation - temperature driven d13C, and others record atmospheric air temperature with d2H or CaCO3d18O. The seasons that are recorded by the different proxies may have a large impact on the reconstructed temperature amplitude, which needs to be discussed.

This is a really good point. After the first sentence of 3.3, we would add

While multiple parameters are sensitive to temperature, because they capture the signal in different parts of the atmosphere-land surface-and cave system, they may record different seasons and therefore potentially also different amplitudes of temperature change. Cave temperatures, which in most settings reflect mean annual temperature, are recorded by TEX86 (Wainer et al, 2011; Matthews et al, 2021; Nehme et al, 2020) and fluid inclusion microthermometry (eg Meckler et al, 2015). Oxygen isotopes measured in calcite or fluid inclusions and dD in fluid inclusions, will reflect the temperature influence on atmospheric processes but biased to the season contributing most to dripwater infiltration, which will vary by setting. For example, the Hungarian caves study by Demeny et al uses winter half year rainfall d2H-temperature relationship for reconstruction while using similar d2H methods, the Wilcox et al study from the Swiss Alps uses an annual rainfall d2H-temperature relationship. All the fluid inclusion oxygen isotope studies and the TEX86 study reconstruct annual average surface temperatures reflected by annual average cave temperatures (Wainer et al, 2011; Matthews et al, 2021; Nehme et al, 2020). The initial carbon isotopic ratio set by soil and vegetation processes may be recorded with seasonal bias if stalagmite deposition has a seasonal bias.

Line 394: At the Jiangjun cave site the amplitude was about 4-5 degrees C, which would correspond to max. 1 permille in calcite d18O. The 2 permille indicated in line 394 as a "temperature effect" is thus not correct. Instead, the additional 1 permille change was explained by a potential bias of the fluid inclusion d18O towards high intensity monsoon rainfall that affected the fabric and incorporation of the fluid inclusions through drip rates.

Thank you. We will make this correction to the text.

Lines 472 – 473: Would it be an idea to use a linear interpolation for the ice volume correction curves, such that you can maintain the original resolution of the speleothem isotope records, but still use an ice-volume correction?

We debated this as well. The age control of speleothems is stronger than the one for the ice volume correction records. Our decision to bin the records takes into account the records with higher age uncertainties.

I wonder if ordering the figures top down according to the timing of the "first response" to the glacial termination would be a better representation of the results. I do believe that it will be easier to read the figure as it would follow the same order as the records are mentioned in the manuscript.

This is what we have tried to do since we found it easier to follow as well.

Line 477: It already starts increasing around 140,000 yrs BP? What is the "starting point" of increasing insolation based on?

In each instance, we have based the 'starting point' at the start of the sharpest rise in insolation. This is at 137,000 years BP for Termination II.

Line 485: This should be coincident with the TII interstadial event, that is actually visible in quite a few EAM records as well as increased runoff in the bay of Bengal (Nilsson-Kerr et al., 2019).

Yes exactly! And as Nilsson-Kerr et al also find, we don't see this in the ISM Bittoo or Xiaobailong cave speleothem records and perhaps a muted signal in the Dongge record but observe a clear signal in the Hulu speleothem record.

5.2. Excellent chapter. The only discussion point you might want to add is that north Europe may be expected to show a cooling in response to freshening and AMOC shutdown, but this is not clear in the northern Europe d18O records.

Precisely because the northern Europe d18O do not consistently show this pattern, we decided not to introduce the "expected" response. One factor may be that the d18Osw influence is only corrected in the North European records for TII, as there is not yet a North Atlantic curve for earlier terminations.

5.4. well done.

Thank you so much!

Chapter 6. Good future directions, well supported by the compiled speleothem Termination records.

Thank you again. We really appreciate the comments, feedback and discussion.

Best wishes,

Jasper Wassenburg

Affolter, S., Häuselmann, A., Fleitmann, D., Edwards, R.L., Cheng, H., Leuenberger, M., 2019. Central Europe temperature constrained by speleothem fluid inclusion water isotopes over the past 14,000 years. Science Advances 5(6), eaav3809. doi.org/10.1126/sciadv.aav3809.

Bajnai, D., Guo, W., Spötl, C., Coplen, T.B., Methner, K., Löffler, N., Krsnik, E., Gischler, E., Hansen, M., Henkel, D., Price, G.D., Raddatz, J., Scholz, D., Fiebig, J., 2020. Dual clumped isotope thermometry resolves kinetic biases in carbonate formation temperatures. Nat. Commun. 11(1), 4005. doi.org/10.1038/s41467-020-17501-0.

Levy, E.J., Vonhof, H.B., Bar-Matthews, M., Martínez-García, A., Ayalon, A., Matthews, A., Silverman, V., Raveh-Rubin, S., Zilberman, T., Yasur, G., Schmitt, M., Haug, G.H., 2023. Weakened AMOC related to cooling and atmospheric circulation shifts in the last interglacial Eastern Mediterranean. Nat. Commun. 14(1), 5180. doi.org/10.1038/s41467-023-40880-z.

Nilsson-Kerr, K., Anand, P., Sexton, P.F., Leng, M.J., Misra, S., Clemens, S.C., Hammond, S.J., 2019. Role of Asian summer monsoon subsystems in the inter-hemispheric progression of deglaciation. Nat. Geosci. 12(4), 290-295. doi.org/10.1038/s41561-019-0319-5.

Wassenburg, J.A., Vonhof, H.B., Cheng, H., Martínez-García, A., Ebner, P.-R., Li, X., Zhang, H., Sha, L., Tian, Y., Edwards, R.L., Fiebig, J., Haug, G.H., 2021. Penultimate deglaciation Asian monsoon response to North Atlantic circulation collapse. Nat. Geosci.(14), 937-941. doi.org/10.1038/s41561-021-00851-9.

Citation: https://doi.org/10.5194/cp-2024-37-RC1

---

## Author Comment (AC2)

RC2: 'Comment on cp-2024-37', Anonymous Referee #2, 09 Aug 2024

This is an interesting manuscript and a good start to stimulate further research on the timing and nature of Terminations. There is no doubt that speleothems have a great potential as they can be dated accurately and precisely with low age uncertainties. The manuscript tries to gather existing speleothem records in order to examine Terminations II, IIIA, III, IV and V in closer detail, with a focus on the sequence of events. I agree with the authors that a comprehensive overview on Terminations in speleothem is currently missing and this overdue. However, I have the feeling that the manuscript was put together quite hastily as the general structure is quite complex and many highly relevant figures are only provided as supplemental information and not in the main text (see comments below). I have the feeling that the authors should try to develop a more concise structure and to present their selection criteria for records more clearly. Furthermore, a more rigid statistical approach is required (see comments below).

Thank you for your comments on the relevance of such a study though we are sorry that you find the structure to be complex, that many figures are in the supplemental material, and that the selection criteria are not quite clear. We understand also that you would like to see a more rigid statistical approach. We will address your comments below and this will hopefully address these concerns.

Some parts on the interpretation of oxygen isotope values in section 2.1 should be moved to section 3. Furthermore, stronger emphasis should be given to the number of dates and sampling resolution of the selected key-records.

Section 2.1 gives only very concise interpretations as a justification for selecting the boundaries of different regions. The actual interpretations are elaborated in Section 3.

We fully understand your concerns regarding dates and sampling resolutions, especially given the nature of this study. That is why we have tried to balance the use of available records with uncertainties generated by low resolution records and those with poor age control. We had 2 choices with record selection, one was to select only the highest quality records at the risk of losing regions from the analysis, and the other was to consider the best records from the different regions with due consideration for their resolution (figures with low resolution records include sample points as markers where uncertainty grey bars are not available) and age control (all U-Th sample points and error bars for every record considered in the manuscript are shown in Figure 2 of the main manuscript and Supplementary Figure 2). We have gone with the second option. In addition to the more regular time series figures, Figure 7 explicitly highlights how the uncertainties in age control may be hindering our understanding of climatic events surrounding Terminations. We also provide uncertainty numbers in the text. Indeed, one of the goals of the manuscript is to highlight where records are available but could do with improvement in resolution and age control.

Many other speleothem records were not really considered in this overview, despite the fact that they could contribute some important additional information on the timing and nature of certain terminations. For instance the timing of the onset of stalagmite growth, e.g. the Sieben Hengste (Switzerland) record covering TII (Luetscher, M., Moseley, G.E., Festi, D., Hof, F., Edwards, R.L., Spötl, C., 2021. A Last Interglacial speleothem record from the Sieben Hengste cave system (Switzerland): Implications for alpine paleovegetation. Quaternary Science Reviews 262.) This record is very well dated and covers Termination II. Within the Alps, the Schafsloch record (Hauselmann et al., QSR, 2016) from Switzerland covering TII is not even mentioned in the text. There are also other speleothem records which could be useful and suited for this review, even if they cover only parts of

a Termination. I think the authors should have done a more comprehensive review of the existing literature. Though some of the records are shown as supplemental figures, it appears that the selection of records in the main text is somewhat arbitrary. Furthermore, the fact that many important figures are shown in the supplemental information doesn't really increase the readability.

The Sieben Hengste record is indeed an excellent one. We have plotted the original record in the supplementary information and the ice-volume corrected record in the main manuscript Figure 4. Based on Reviewer 1's comments, we have also explicitly stated this in subsection 2.2. This may be confusion created because it was not plotted in Figures 1 and 2 or shown in Table 1. We are sorry about this. The figure was getting too crowded to show the Abaliget, Sieben Hengste and Schneckenloch records. Therefore, we show the Abaliget record which covers the whole Termination with reasonable resolution and age control (as per our record selection criteria) in Figure 2 along with its age control, and do the same for the Sieben Hengste and Schneckenloch records in the Supplementary Information. The Schafsloch record is a really nice one as well and one of the first covering this time period from the region. We were already showing the Abaliget, the Sieben Hengste and the Schneckenloch records from this region in the manuscript. The Schafsloch record is of excellent quality but covers a shorter time period than the 3 other records already in the manuscript, that is why, as per our sample selection criteria, this record has not been shown.

We believe that we have been as comprehensive as reasonably possible for this manuscript. SISAL is the largest speleothem database, and as mentioned in the manuscript, the database was built parallel to working on this project so that we have made every effort to track down speleothem records covering Termination TII through TV. We have mined the database systematically for any record within the Termination time periods and selected the most suitable ones (per our criteria given in the Methods section) for further discussion in the manuscript and Supplementary information.

The selection criteria of records are certainly arbitrary in the sense that it doesn't follow rigid criteria of a particular number of U-Th ages or a particular resolution. As we mention in the previous comment, we did this so that we could consider more records from more regions with due consideration for linked uncertainties.

We spent quiet a lot of time debating which figures should go in the main manuscript and which should go in the supplementary information. We would be happy to add more figures to the main manuscript perhaps also aided by the Editor's suggestions.

It remains unclear to what extent different age models (COPRA/Bchron/Stalage etc.) have an effect on the timing of Terminations and it would be useful to show the effects on 2-3 records in the main text. If the effects are minimal, then one can exclude at least on potential source of uncertainty.

This is a really good point, and as you say, merits more work. For example, Figure 2 in Perez-Mejias et al, 2017 highlights the difference in modeled ages based on two age depth models, OxCal and StalAge, even in records where the uranium-thorium ages have low uncertainties. In this case the authors have elected to use a mixture of age-depth model methods for creating the final age model. It is for nuances like this that we choose to use the author generated age models as a priority as long as the authors have provided uncertainty data.

There is a non-trivial amount of analysis to be done using a function like change point and taking into consideration the uncertainties from all the age depth model ensembles. This work could also

consider millennial events surrounding Terminations. This is work we hope to do in the future and has been listed in the future work section. That analysis is beyond the scope of this manuscript. In this manuscript, we take the first steps i.e. (i) plotting the U-Th sample points with their measured uncertainties, (ii) indicating some low-resolution records and (iii) showing uncertainties resulting from age-depth models on the figure (iv) author-generated or the same age models wherever possible to try and minimise uncertainties resulting from the use of different age-depth models.

A stronger consideration of carbon isotope records would be also useful, particularly for speleothem records from temperate regions where vegetation and soil microbial activity are highly dependent on temperature and rainfall. The full potential of the speleothem isotope records is not exploited

We agree with this comment. d13C is really an under-utilised proxy in speleothems. The limitation in including further records in these figures has been the availability of trace element or calcium isotope data to evaluate the PCP effects. We cite some excellent recent work making the most use of this proxy:

Genty et al, 2006

Lechleitner et al, 2021

Stoll et al, 2023

And we will add a point to the future work section as follows:

**Speleothem d13C proxy records, particularly from temperate regions, are showing great promise in reconstructing past changes in temperature and rainfall when coupled with other proxies such as Mg/Ca and dCa. Such records from the particularly data dense Northern temperate regions would add great value to research on Terminations.**

Is section 2.3 really necessary as no 13C record is shown in the current version of the manuscript.

We are sorry for this confusion. The d13C records are mentioned in section 2.3 and the degassing corrected version of the records have directly been plotted in Figures 4 and 6. The correction itself is shown in Supplementary Figure 5 which is creating this confusion. We can move Supplementary Figure 5 to the main manuscript into Section 2.3. The Termination II La Vallina cave Garth speleothem record and the Termination III Ejulve cave Artemisa speleothem record d13C records have been discussed in the manuscript.

The Figure 7 is not always correct. For instance, for TII, there is only one temperature increase in Europe, whereas the text states "final step of temperature increase in Europe and North America" (lines 644-645. Please make sure that Figure 7 is indeed conform with the main text. Furthermore, please make clear how the amplitude of the change was calculated. Ice volume correction applied. In this figure, one could also display insolation forcing to reveal the phasing more clearly.

Thank you for spotting that. We have added the final step of temperature increase in Europe for Termination II now.

Figure 7 shows the amplitude as given in the individual Termination Figures 4, 5 and 6. And Figure 7 captures the climate changes from these figures that aid in the discussion of section 5.2. We have clarified this in the Figure caption now:

Figure 7: Sequence of **selected** global climatic events over Terminations. Ages and chronological uncertainties are represented on the X-axes. Amplitude of oxygen isotope changes that reflect the climatic events in speleothem records are plotted on the Y axes. **The amplitudes are taken from Figures 4, 5 and 6 for the respective Terminations.** The dashed line shows the start of insolation increase. [precip = precipitation; temp = temperature; N Eu = North Europe; S Eu = South Europe; N Am = North America; C As = Central Asia; ISM = Indian Summer Monsoon; EASM = East Asian Summer Monsoon; SE Asia = Southeast Asia; S Am = South America]

That's a really good idea regarding insolation! We will add that curve to the figure.

Thank you for your suggestions on this figure. These changes will make the figure much clearer!

[Figure]

The effects of the "ice volume-correction" should be also shown more clearly as this is an important aspect as ice-volume corrections can affect the overall structure of Terminations.

This may again be a case where we have moved some of the figures to supplementary information. The different sea level curves and ice volume effects of the different Terminations have been plotted in Supplementary Figure 3. The impact of the different ice volume corrections on all the records used in the main manuscript have been shown in Supplementary Figure 4. In the main manuscript itself, we have opted to show only the main record figures. So that the uncorrected records are shown in Figure 2. And the ice-volume corrected records, where the corrections do indeed make a difference to the structure of the Termination, are shown in Figure 4 with the Y-axis labelled d18Ocorr.

Specific Comments:

Section 3.1 Records of surface ocean freshening: In this section, the Corchia Cave record should be also mentioned.

A careful comparison of the d18Osw (from foraminiferal d18O and Mg/Ca) and the d18O of Corchia stalagmites over the last deglaciation has shown that d18Osw is not the dominant control of Corchia d18O (Stoll et al., 2022), so we have not included it in this section. Thus, we retain the section as is and discuss the regional signal from Corchia.

Lines 120-121: What is the specific rationale behind the use of single records and not composite records? Composite records are considered to be more robust than individual records.
Citation: https://doi.org/10.5194/cp-2024-37-RC2

Composite records and stacks have been known to increase the robustness of records by strengthening regional signals versus drip-site specific noise and by expanding chronological control. We elected to use single records because they gave us more information on age control and measured d18O values without having to account for modifications made to either during the process of creating composites.

---

## Editor Decision (ED1)

**T paper**

This is a review paper presenting speleothem records covering terminations (T) with the aim to (i) synthesize the available speleothem records covering T II, TIIIA, TIII, TIV and TV, (ii) description of their proxy quality, and (iii) Evaluation of the leads and lags of the records.

The impact of a review paper depends on a well-organized structure and clear figures and tables. Unfortunately, the revised version is lacking clarity and a clear structure. For instance, the selection criteria for the stalagmites remain unclear, which is also related to the fact that the quality of the main table is still very bad.

I think the criteria are set out sufficiently. I don't necessarily agree with all the criteria (e.g. combining multiple speleothem records is designed to improve timings/comparisons, etc., not make them worse), but I appreciate the need for some sort of consistency when screening records. You should consider removing T-IV and T-V altogether – the two records from each (from SE Asia/EASM only) add nothing, and you cannot compare records between regions. The very imprecise chronology of the Green Cathedral record (T-V) also does not help the cause.

The table definitely needs a makeover. Please consider presenting only the most essential metadata information for the main text and keeping the current expanded version for the supplementary data. A landscape (rather than portrait) format might work best for the supplementary (full) version (or both).

This is also true for some of the figures, particularly for Figure 2, which would stretch over three pages in a publication.

For Figure 2, please:
- in your textbox descriptions of each plot, just use the cave names to avoid clutter. Add the description of what each proxy series represents in the caption. Also, the textboxes are placed over the time series in places – please avoid this.
- best not to combine blue and green (for readers with colour resolution problems). See https://www.nature.com/articles/nmeth.1618
- offset the U-Th dates/error bars vertically in a better way so that symbols/bars do not superimpose one another.
- provide the full description of the insolation metric of Tzedakis et al. 2017 for your first plot (and x-ref to this caption in the following plots) and specify the unit of measurement. Also, offset the maximum insolation value in each case from the upper x-axis by expanding the y-axis range, and consider using the same y-axis range for each plot so amplitude differences in the metric between terminations can be compared more readily.
- be more generous with information in the caption. A figure and caption should be 100% self-contained as much as possible (i.e. all the essential information for interpretation should be provided with the need to consult the main text).

The major weakness, however, is the discussion about similarities and differences between the Terminations (chapter 5), which should be the most important part of the review. The absolute timing and nature of Terminations in the speleothems is key to understand the interplay between external (orbital parameters) and internal (e.g., glacial boundary conditions, ocean circulation, carbon dioxide) climate forcing mechanisms. These important aspects are not really discussed in paragraphs 5.1 and 5.2, which are very short compared to the preceding paragraphs. This surprises me, since the authors state in the introduction that "we evaluate the whether there are consistent leads and lags in the manifestation of terminations across different aspects of the climate systems and different regions as tracked by speleothem proxy records.".

Given the authors' intent, it is reasonable to expect the Discussion section to be central to the paper, so I agree with the reviewer that this part needs more work and better organisation. For example, I do not see the point in comparing NISA for T-II with Sofular for T-III (lines 613 onwards). The main features of similarity and difference for how speleothems are recording local/regional climate through each termination, and why this might be the case, should be the focus. Do the apparent leads and lags makes sense? Are they confounded by uncertainties in the age models being used (not to mention the fact that some papers use more than one age-model algorithm)? It is from these observations that deficiencies in the current state of knowledge can be highlighted. Therefore, I urge you to carrying out the most plausible and meaningful comparisons between the speleothem records for each termination (still looking at regional differences), bearing in mind the dating constraints.

Chapter 5 is rather poorly organized and it is very difficult to follow their arguments and selection criteria. To give an example: the sequence of events for Termination II in speleothems is shown in Figure 4, whereas only insolation changes are displayed as one external forcing factor. What about other key-forcing factors, such as AMOC, IRD or carbon dioxide? Changes in the intensity of the AMOC, for instance, are responsible for cold/dry snaps during Terminations (YD-like events), and they should be discussed in greater detail as they are one important aspect of almost all Terminations.

Whilst this was not requested in the first round of reviews, I believe adding additional information in the figures is useful for contextualising the comparisons you are making. The difficulty of assigning chronologies to ice-core and marine records is not trivial so you should make this clear when introducing these time series in the text. For example, time series from the marine record would be useful – e.g. a benthic and SST from the same core (=same chronology) from the Iberian margin).

In Figure 7, the x-axis ranges should be extended to capture the complete error bars, and I would recommend either excluding the 'start of insolation' indicator, or apply it to the first derivative of the insolation metric, which appears as though it will show an uptick in insolation as the termination kicks in.

Also, here and through the paper, please be consistent with expressing time (both age and duration): it should be expressed in ka or kyr respectively (with 1 decimal place where necessary, but not more), and not in years (e.g. lines 629 and 643).

Figure 7 is only of limited use as the authors focus solely on oxygen isotope magnitudes, whereas growth phases (important for some alpine sites and related to cold snaps) are not included. In addition, the amplitude of the few existing paleotemperature estimates is also not included. The sequence of events in Figure 7 is therefore incomplete and rather selective.

I agree it is useful to include in Fig. 7 growth phases and temperature estimates as suggested by the reviewer (incorporate these into the main text) but urging caution regarding interpretation of any single speleothem records of growth/interruption.

More importantly, it is still unclear how the timings, durations and amplitudes of these events/intervals were calculated.

Please specify these in the text or figure caption where applicable.

Not to mention that the chronologies of the records in Figure 4 are based on different chronology building approaches.

I agree that this is an issue, but it cannot be dealt with in this paper (it's a separate and long overdue study itself).

Though one could find some information in the table, some of the records shown in figure 4 are not included in Table 1 (e.g., Siebenhengste, Schneckenloch, Diamante).

Please rectify

Furthermore, my initial concern regarding a more scientific and statistically sound approach to calculate ages for the onset/end of the termination and timing of dry/cold or warm/wet phases were not addressed in the revised manuscript. This is still a major weakness, particularly because the authors give sometimes extremely precise age estimates for some terminations, such as "~6985 years from the initial weak monsoon event to monsoon recovery".

See comment above re expression of time in ka/kyr, and add uncertainties in quadrature when specifying durations (although some age models seem to underestimate interpolation uncertainties!).

Overall, I think that the current manuscript is not suited for publication in CoP as it reads more like an initial draft. In my first review, I had hopes that the authors would revise their manuscript thoroughly, but they missed this opportunity, leaving me with no choice but to reject it. However, a review on the timing of terminations in different speleothem records is overdue and I would encourage the authors to revise and resubmit their manuscript to CoP.

I agree – with a little more work, this ms can make a useful contribution

---

## Author Response (AR2)

Terminations paper

Responses to Reviewer and Editor comments

In grey – Reviewer comments

In red – Editor comments

In blue – author comments

Lines 69-72: dragon sentence :). Consider to rephrase to: "While such tuning approaches are promising, confidently applying speleothem chronologies to multiple Terminations and millennial events during Terminations requires robust proxy interpretations across multiple climate archives. Ideally, modelling studies point to common drivers of proxy variability.

Thank you for this suggestion. We have made the change.

Lines 348-349: The TEX86 proxy is not used in these references.

You are correct. We have now modified the sub-heading to reflect that this sub-section only refers to speleothem carbonate oxygen and carbon isotope records rather than TEX86 or fluid inclusion records which are instead referred to in sub-section 3.3.

This is a review paper presenting speleothem records covering terminations (T) with the aim to (i) synthesize the available speleothem records covering T II, TIIIA, TIII, TIV and TV, (ii) description of their proxy quality, and (iii) Evaluation of the leads and lags of the records. The impact of a review paper depends on a well-organized structure and clear figures and tables. Unfortunately, the revised version is lacking clarity and a clear structure. For instance, the selection criteria for the stalagmites remain unclear, which is also related to the fact that the quality of the main table is still very bad.

I think the criteria are set out sufficiently. I don't necessarily agree with all the criteria (e.g. combining multiple speleothem records is designed to improve timings/comparisons, etc., not make them worse), but I appreciate the need for some sort of consistency when screening records. You should consider removing T-IV and T-V altogether – the two records from each (from SE Asia/EASM only) add nothing, and you cannot compare records between regions. The very imprecise chronology of the Green Cathedral record (T-V) also does not help the cause. The table definitely needs a makeover. Please consider presenting only the most essential metadata information for the main text and keeping the current expanded version for the supplementary data. A landscape (rather than portrait) format might work best for the supplementary (full) version (or both).

We recognize the challenges in the table and we have made a full revision.

Regarding T-IV and TV, we would like to retain them in the manuscript to highlight the need for records from these Terminations and these regions and to lay a template for comparisons across all terminations as additional records on these Terminations will become available (e.g. preprints and conferences highlight upcoming works from Torner et al covering T-IV from the Iberian region (https://doi.org/10.5194/cp-21-465-2025), upcoming works on the SE Asian speleothem fluid inclusion records from Nele Meckler's group

(https://doi.org/10.5194/egusphere-egu24-15117) and the Drysdale group presentations at the Karst Records conference (Ulasi et al, Pollard et al: https://airdrive.eventsair.com/eventsairwesteuprod/production-uctcmc-public/b00000003b3d4576996c7d71aefad4da).

This is also true for some of the figures, particularly for Figure 2, which would stretch over three pages in a publication.

For Figure 2, please: in your textbox descriptions of each plot, just use the cave names to avoid clutter. Add the description of what each proxy series represents in the caption. Also, the textboxes are placed over the time series in places – please avoid this. best not to combine blue and green (for readers with colour resolution problems). See https://www.nature.com/articles/nmeth.1618 offset the U-Th dates/error bars vertically in a better way so that symbols/bars do not superimpose one another. provide the full description of the insolation metric of Tzedakis et al. 2017 for your first plot (and x-ref to this caption in the following plots) and specify the unit of measurement. Also, offset the maximum insolation value in each case from the upper x-axis by expanding the y-axis range, and consider using the same y-axis range for each plot so amplitude differences in the metric between terminations can be compared more readily. be more generous with information in the caption. A figure and caption should be 100% self-contained as much as possible (i.e. all the essential information for interpretation should be provided with the need to consult the main text).

I have been working on improving these figures as we have presented at conferences. I'm sorry that they were rather unclear in the publication. The figures have been improved.

- Plots include only cave names,
- textboxes and U-Th ages and errors are offset from time series,
- colours have been changed,
- the insolation metric has been described more fully and includes the units of measurement,
- the insolation values on Y axis have been set to the same range across the different Terminations,
- more full descriptions have been given below figures.

The major weakness, however, is the discussion about similarities and differences between the Terminations (chapter 5), which should be the most important part of the review. The absolute timing and nature of Terminations in the speleothems is key to understand the interplay between external (orbital parameters) and internal (e.g., glacial boundary conditions, ocean circulation, carbon dioxide) climate forcing mechanisms. These important aspects are not really discussed in paragraphs 5.1 and 5.2, which are very short compared to the preceding paragraphs. This surprises me, since the authors state in the introduction that "we evaluate the whether there are consistent leads and lags in the manifestation of terminations across different aspects of the climate systems and different regions as tracked by speleothem proxy records.".

Given the authors' intent, it is reasonable to expect the Discussion section to be central to the paper, so I agree with the reviewer that this part needs more work and better organisation. For example, I do not see the point in comparing NISA for T-II with Sofular for T-III (lines 613

onwards). The main features of similarity and difference for how speleothems are recording local/regional climate through each termination, and why this might be the case, should be the focus. Do the apparent leads and lags makes sense? Are they confounded by uncertainties in the age models being used (not to mention the fact that some papers use more than one age-model algorithm)? It is from these observations that deficiencies in the current state of knowledge can be highlighted. Therefore, I urge you to carrying out the most plausible and meaningful comparisons between the speleothem records for each termination (still looking at regional differences), bearing in mind the dating constraints.

We have clarified in the introduction the purpose of the paper (final paragraph of introduction):

In this review paper we identify how currently available speleothem records constrain the timing of climate events across different regions within a given termination, and describe the similarities and differences in the sequences of a processes across different Terminations.

Additionally, we retitled heading 5.1 and have rewritten the first main paragraph to emphasize the comparison of a similar process (meltwater release) among TII and TIIIA detected in two speleothems (NISA and Sofular).

Both the NISA and Sofular $\delta^{18}O$ are inferred to track the release of isotopically light meltwater from Northern Hemisphere ice sheets, albeit in different regions, and in both TII and TIIIA, meltwater release happens similarly early in the Termination prior to the warming in North Europe, and recovery of the EASM. The differences in magnitude of the $\delta^{18}O$ signals in NISA and Sofular is a manifestation of the different sources of moisture for the two records rather than a reflection of the differences between Terminations, since the NISA records the ~2‰ freshening of the North Atlantic is thought to have altered the source composition by ~2‰ (Stoll et al., 2022) whereas a combination of factors including the inflow of isotopically depleted Caspian Sea waters resulting from Eurasian ice sheet melt, diversion of rivers, higher river runoff coefficients and reduced evaporation to the Black Sea during Terminations contributes to the strongly depleted $\delta^{18}O$ values of the Sofular cave record (Badertscher et al., 2011 and references therein).

Chapter 5 is rather poorly organized and it is very difficult to follow their arguments and selection criteria. To give an example: the sequence of events for Termination II in speleothems is shown in Figure 4, whereas only insolation changes are displayed as one external forcing factor. What about other key-forcing factors, such as AMOC, IRD or carbon dioxide? Changes in the intensity of the AMOC, for instance, are responsible for cold/dry snaps during Terminations (YD-like events), and they should be discussed in greater detail as they are one important aspect of almost all Terminations.

Whilst this was not requested in the first round of reviews, I believe adding additional information in the figures is useful for contextualising the comparisons you are making. The difficulty of assigning chronologies to ice-core and marine records is not trivial so you should make this clear when introducing these time series in the text. For example, time series from the marine record would be useful – e.g. a benthic and SST from the same core (=same chronology) from the Iberian margin.

We have added a new figure to illustrate an example of speleothem age tuning of marine records. We have expanded the discussion of this in the first paragraph of section 5.3.

Tzedakis et al., (2018) match the relative abundance of temperate pollen in marine sediment records on the Iberian margin pollen with Corchia (Italy) speleothem $\delta^{18}O$, assuming a common rainfall amount signal dominates both records (Figure 8). This provides an independent, speleothem tuned chronology for other proxies recorded in the Iberian margin sequence, such as the benthic $d^{18}O$, and reveals how the chronology of events in the marine sediment core would be shifted earlier by several thousand years compared to a marine age model based on tuning benthic $d^{18}O$ to stacks (e.g. Spratt and Lisiecki, 2016).

This last sentence clarifies precisely why we have not included on figures the curves of climatic parameters from marine sediment or ice core records which are not well tied to absolute chronology – because the chronologies of these records can be offset by many thousands of years, so it is not possible to describe the sequence of timing across a termination from records derived from different archives.  Thus, we refrain from adding comparative records from non-tuned climate parameters from marine or ice core records

In Figure 7, the x-axis ranges should be extended to capture the complete error bars, and I would recommend either excluding the 'start of insolation' indicator, or apply it to the first derivative of the insolation metric, which appears as though it will show an uptick in insolation as the termination kicks in. Also, here and through the paper, please be consistent with expressing time (both age and duration): it should be expressed in ka or kyr respectively (with 1 decimal place where necessary, but not more), and not in years (e.g. lines 629 and 643).

-   We have increased the horizontal size of this figure to better represent the information but expanding the Y-axis to cover complete error bars makes the figure unwieldy.
-   We have removed the start of insolation metric and instead simply show the insolation curve on the figure with the same Y-range across all Terminations to enable comparison.
-   We have expressed time everywhere in ky BP or ka now and restricted age expression to one decimal place.

Figure 7 is only of limited use as the authors focus solely on oxygen isotope magnitudes, whereas growth phases (important for some alpine sites and related to cold snaps) are not included. In addition, the amplitude of the few existing paleotemperature estimates is also not included. The sequence of events in Figure 7 is therefore incomplete and rather selective.

I agree it is useful to include in Fig. 7 growth phases and temperature estimates as suggested by the reviewer (incorporate these into the main text) but urging caution regarding interpretation of any single speleothem records of growth/interruption.

We fully agree with the editor that extreme caution would be necessary to make robust interpretations of climate from speleothem growth phase/interruption.  For example, work in the recent paper on growth phases in the British isles by Panitz et al (https://doi.org/10.5194/cp-21-261-2025)  really exemplifies both the power and the complexity of using growth phases to derive paleoclimatic information. For this reason, we propose to not overextend an already long paper by introducing an additional proxy (growth phases/hiatus).  Rather, we hope that future projects and collaborations can take on the task of integrating information from growth phases and growth stops in a transparent and robust way to provide complementary data.  Regarding the proxies discussed in this paper and in Figure 7, we compare the oxygen isotope records of timing of temperature changes and extend these

by describing in the discussion the absolute temperature estimates from Hungarian caves and from Schrattenkarst.

More importantly, it is still unclear how the timings, durations and amplitudes of these events/intervals were calculated.

Please specify these in the text or figure caption where applicable.

Ah that's a good point. I'm sorry. I have expanded on this in the methods section in sub-section 2.1.

Not to mention that the chronologies of the records in Figure 4 are based on different chronology building approaches.

I agree that this is an issue, but it cannot be dealt with in this paper (it's a separate and long overdue study itself).

Though one could find some information in the table, some of the records shown in figure 4 are not included in Table 1 (e.g., Siebenhengste, Schneckenloch, Diamante).

Please rectify

We have elaborated on the source of the information and corrections further in the figure captions. I'm sorry that this is causing confusion.

Furthermore, my initial concern regarding a more scientific and statistically sound approach to calculate ages for the onset/end of the termination and timing of dry/cold or warm/wet phases were not addressed in the revised manuscript. This is still a major weakness, particularly because the authors give sometimes extremely precise age estimates for some terminations, such as "~6985 years from the initial weak monsoon event to monsoon recovery".

See comment above re expression of time in ka/kyr, and add uncertainties in quadrature when specifying durations (although some age models seem to underestimate interpolation uncertainties!).

I'm sorry about this. We have made this correction.

Overall, I think that the current manuscript is not suited for publication in CoP as it reads more like an initial draft. In my first review, I had hopes that the authors would revise their manuscript thoroughly, but they missed this opportunity, leaving me with no choice but to

reject it. However, a review on the timing of terminations in different speleothem records is overdue and I would encourage the authors to revise and resubmit their manuscript to CoP.

I agree – with a little more work, this ms can make a useful contribution

We appreciate the clarification on additional aspects of the paper which require further revisions and have diligently followed through on these points.

---

## Author Response (AR3)

Dear Editor,

We are sorry for the typos particularly in cave names. We have double-checked and corrected spellings across the files.

Best wishes
Authors